# A machine learning approach for evaluating Southern Ocean cloud-radiative biases in a global atmosphere model

Sonya L. Fiddes[1,2], Marc D. Mallet[1], Alain Protat[3,1], Matthew T. Woodhouse[2,1], Simon P. Alexander[4,1], and Kalli Furtado[5,*]

[1]Australian Antarctic Program Partnership, Institute of Marine and Antarctic Studies, University of Tasmania, Hobart, Australia
[2]Climate Science Centre, Oceans and Atmosphere, Commonwealth Scientific and Industrial Research Organisation, Aspendale, Australia
[3]Bureau of Meteorology, Melbourne, Australia
[4]Australian Antarctic Division, Hobart, Australia
[5]Met Office, Exeter, United Kingdom
[*]Now at Centre for Climate Research, Meteorological Service Singapore

**Correspondence:** Sonya L. Fiddes (sonya.fiddes@utas.edu.au)

**Abstract.** The evaluation and quantification of Southern Ocean cloud-radiation interactions simulated by climate models is essential in understanding the sources and magnitude of the radiative bias that persists in climate models for this region. To date, most evaluation methods focus on specific synoptic or cloud type conditions that do not consider the entirety of the Southern Oceans cloud regimes at once. Furthermore, it is difficult to directly quantify the complex and non-linear role that different cloud properties have on modulating cloud radiative effect. In this study, we present a new method of model evaluation, using machine learning, that can at once identify complexities within a system and individual contributions.

To do this, we use an XGBoost model to predict the radiative bias within a nudged version of the Australian Community Climate and Earth System Simulator – Atmosphere-only Model, using cloud property biases as predictive features. We find that the XGBoost model can explain up to 55% of the radiative bias from these cloud properties alone. We then apply SHapley Additive exPlanations feature importance analysis to quantify the role each cloud property bias plays in predicting the radiative bias. We find that biases in liquid water path is the largest contributor to the cloud radiative bias over the Southern Ocean, though important regional and cloud-type dependencies exist. We then test the usefulness of this method in evaluating model perturbations and find that it can clearly identify complex responses, including cloud property and cloud-type compensating errors.

## 1 Introduction

The Southern Ocean (SO) shortwave cloud radiative bias is a well documented problem in global climate models (Bodas-Salcedo et al., 2014; Schuddeboom and McDonald, 2021) as well as some numerical weather prediction models (Protat et al., 2017; McFarquhar et al., 2021). The bias is characterised by too much shortwave radiation reaching the surface of the ocean, and not enough being reflected by clouds back out to space. Significant work has been done to identify the cause of this

model bias. Haynes et al. (2011) and Bodas-Salcedo et al. (2016) have shown that a large part of this bias can be attributed to the inability of models to simulate super-cooled liquid water clouds in the SO, in particular in cold sectors of extra-tropical cyclones. A number of observational studies have shown the prevalence of super-cooled liquid water clouds over the SO (Huang et al., 2012; Chubb et al., 2013; Mace and Protat, 2018). The prevalence of super-cooled liquid water clouds is attributed to the pristine conditions found in the region, removed from the sources of terrestrial (eg. dust/biomass burning) and anthropogenic (eg. black carbon) aerosol (although these aerosol can occasionally intrude into the region). The lack of these particular aerosol, which contribute large sources of ice-nucleating particles (INP), limits the ability of cloud droplets to freeze, resulting in liquid clouds at temperatures well below zero. Currently, many climate and weather models do not take into account the pristine composition of the SO atmosphere, assuming, like over much of the world, that INP are available to help freeze cloud droplets, resulting in too many ice-phase clouds, which allow too much shortwave radiation to reach the surface of the ocean (Vergara-Temprado et al., 2018; McCluskey et al., 2023).

Numerous studies have attempted to address the shortwave cloud radiative bias via cloud phase parameterisations, including, but not limited to, treatment of ice nucleating temperatures (Furtado and Field, 2017; Varma et al., 2020, 2021), ice crystal shapes (Varma et al., 2020) and growth rates (Furtado et al., 2016), ice nucleating particle (INP) number concentrations (Vignon et al., 2021) and sources (Vergara-Temprado et al., 2017, 2018), convective detrainment temperatures (Kay et al., 2016) and more. Invariably, many of these studies find that altering parameters for specific SO conditions results in changes in model performance over other parts of the climate system for better or worse (as explored in Kay et al., 2016; Furtado et al., 2016; Varma et al., 2020, 2021). This outcome is particularly important given the range of recent literature highlighting a latitudinal dependence of cloud properties in the Southern Ocean, which has been attributed by some to differences in aerosol properties (McCoy et al., 2015; Humphries et al., 2021; Mace et al., 2021b, a).

Many model evaluation techniques used to diagnose this problem rely on satellite observations and hence require a satellite simulator such as the Cloud Feedback Model Intercomparison Project (CFMIP) Observation Simulator Package (COSP) described in Bodas-Salcedo et al. (2011). More recently, ground-based lidar simulators are also available for more accurate model evaluation from ceilometers and lidars on the surface (eg. Kuma et al., 2020). In conjunction with such simulators, most studies have separated their data into specific conditions, for example, by isolating particular synoptic situations (eg. Field and Wood, 2007; Bodas-Salcedo et al., 2016) or by cloud regimes (or 'weather states') (eg. Williams and Webb, 2009; Tselioudis et al., 2013; Oreopoulos et al., 2014; Mason et al., 2015; Oreopoulos et al., 2016; McDonald et al., 2016; Leinonen et al., 2016; Schuddeboom et al., 2018; Tselioudis et al., 2021; Fiddes et al., 2022). By isolating particular conditions, the specific microphysical causes of the radiative bias that are relevant to that condition can be diagnosed. However that cause may not perhaps be relevant to other conditions. Additionally, as Fiddes et al. (2022) (a companion paper to this study; herein F22) found, the model being examined in this study rarely simulates cloud regimes correctly, which may reduce the usefulness of the cloud regime approach and calls for a different method of evaluation. Without the use of synoptic typing or cloud regimes, calculating zonal means is a popular way of diagnosing model biases at a macro-scale. However, this method severely limits the microphysical inferences that can be made.

Another way to evaluate and in some instances tune models is to explore parameter uncertainty (Lee et al., 2013; Regayre et al., 2020, 2023). In these cases the parameter space (the range of plausible values) and their impacts in global climate models are emulated with more simplified statistical models. This allows re-sampling over a range of multi-parameter values many times over what is possible with physically driven models. From these large samples, the uncertainty attributed to particular parameters can be identified and the best combination of parameter values can be constrained based on comparisons with observations. These methods present a powerful way of reducing uncertainty of climate models within known and quantified parameters and physical mechanisms.

The evaluation techniques presented in the literature are important methods in understanding model biases and have been shown to be useful in testing and tuning new parameterisations. However, these techniques are often limited to using human ability to discern complex physical processes, interactions and patterns to diagnose the drivers of biases. We suggest that utilising machine learning and associated feature importance metrics can enhance pattern recognition, aid our ability to assess non-linearity and collinearity and shed new light on our understanding of the underlying causes of the biases across multiple conditions.

Increased computing power as well as increased data availability now means that machine learning techniques are a useful tool to further understand and predict climate and weather problems, especially in relation to clouds (Beucler et al., 2021). Currently, applications of machine learning in climate science are limited for a number of reasons, including the relatively recent advances in the field with respect to both methods and accessibility of computing resources, the difficulty in applying often non-perfect or very limited data sets to a problem and the fact that physical understanding in climate science is often more important than model accuracy (Beucler et al., 2021). Current applications include (but are not limited to): predicting a particular field, such as low-marine clouds (Fuchs et al., 2018), liquid water path (Zipfel et al., 2022), radiation (Fan et al., 2018; Mallet et al., 2023); improving retrievals for remote sensing (Yan et al., 2020); downscaling of coarse resolution data (Vandal et al., 2019); improving subgrid-scale parameterisations (Rasp et al., 2018); or classification problems (Zhang et al., 2019).

Of particular note is the application of machine learning to climate emulation, i.e. emulating the global response of complex climate models, as outlined in Watson-Parris et al. (2022). Climate emulation has typically used simple models to estimate what the response of the climate (usually temperature) may be to changes in forcings. These models tend to not capture spatially varying and non-linear processes well, whereas machine learning has been shown to do well in this space, but has been challenged by a lack of data for training purposes. Watson-Parris et al. (2022) have now provided a dataset and some initial machine learning frameworks designed specifically for training models for this application, which may provide a new way to determine possible climate responses to changes in forcings, beyond that of the temperature.

To our knowledge, no study has yet applied machine learning methods to understanding biases in climate models. We now believe that current methods for regression problems combined with feature importance metrics may provide useful insight into how biases are occurring. Feature importance metrics aim to provide quantification of how each predictive feature in a problem (regressive or categorical) has contributed to the final outcome. Commonly, feature importance metrics have been considered misleading due to their inability to take into account dependencies between predictive features (Hooker et al., 2021). However,

the recent advances in this space, including the development of the SHAP (SHapley Additive exPlanations) feature importance,
mean that we can better take into account these dependencies.

SHAP analysis builds on Shapley Values, a method originally derived for game theory applications to identify how important one player in a team was to the outcome of the game (Shapley, 1953; Lundberg and Lee, 2017). SHAP feature importance can be used to assess individual, accumulative and interacting feature importance, taking into account collinearities. Furthermore, SHAP analysis is model agnostic and is considered a powerful tool in feature analysis.

In this study, we combine SHAP feature analysis with a regression model to evaluate and understand cloud radiative model biases for the first time. We test if we can perform such an evaluation in a more holistic manner than in previous studies, considering all conditions at once, rather than specific cloud or synoptic regimes. To do this, rather than isolating a particular regime and then examining the particular biases in cloud properties associated with it, biases in cloud microphysical properties are used to predict the bias in the cloud radiative effect. We then apply SHAP feature importance to understand the primary
drivers of the cloud radiative bias at any point in space or time. We hypothesis that that this method can provide new insight into the cloud-radiative bias problem and may be useful tool when it comes to model sensitivity testing.

## 2    Data and methods

### 2.1    ACCESS-AM2 model and observational products

The Australian Community Climate and Earth System Simulator (ACCESS) - Atmosphere-only Model Version 2 (AM2) model
is used in this work (Bodman et al., 2020). ACCESS-AM2 uses the same atmospheric set-up as that of the Coupled Model Intercomparison Project (CMIP) ACCESS-Coupled Model 2 (CM2) (Bi et al., 2020) submission for the AMIP (atmosphere only model intercomparison project) design (Eyring et al., 2016), but with prescribed sea surface temperatures and sea-ice concentrations. While we will provide key details here, a full description of the exact model set-up can be found in F22 (Fiddes et al., 2022). The atmospheric model is the Unified Model (UM) vn10.6, GA7.1 Walters et al. (2019), used in conjunction with
the Community Atmosphere Biosphere Land Exchange (CABLE) version 2.5 land surface scheme (also described in Bi et al., 2020) and the GLOMAP-mode (GLObal Model of Aerosol Processes) aerosol microphysical scheme (Mann et al., 2010; Ma et al., 2012). Importantly, for this work we have the COSP simulator switched on (Bodas-Salcedo et al., 2011), in this case for the the Moderate Resolution Imaging Spectroradiometer (MODIS) satellite, to allow for sensible comparison between satellite fields and the model.

We have run the model from 2014-2019, discarding the year 2014 as spin-up. The European Centre for Medium-range Weather Forecasting (ECMWF) Reanalysis 5 (ERA5) product (Hersbach et al., 2020) is used to nudge the model every three hours, using the the horizontal wind and temperature above the boundary layer. The ACCESS model runs at 1.25x1.875 degree horizontal resolution with 85 vertical levels, and for this work we have output daily means.

We use two satellite products in this work: cloud properties from MODIS Combined Aqua/Terra, Level 3 daily, 1x1 degree
grid, Collection 6.1, COSP product (MCD06COSP_D3_MODIS) derived specifically for CMIP6 (Pincus et al., 2012; Platnick et al., 2017; Hubanks et al., 2020) and radiation fields from the Clouds and the Earth's Radiant Energy System (CERES)

Syn1Deg product (Doelling et al., 2013, 2016). Both these products are available at daily mean time-scales and have been regridded to match the ACCESS-AM2 horizontal grid. How these products have been prepared is fully described in F22, which has used the exact data set as this current work. F22 includes discussion about the satellite products strengths and limitations, quality (including successful pixel retrieval), past evaluation and processing.

We use the outgoing top of atmosphere (TOA) shortwave (SW) cloud radiative effect (CRE) ($SWCRE_{TOA}$) for this work. We have defined the $SWCRE_{TOA}$ as the difference between the clear-sky radiation and the all-sky radiation fields (for both the model and the satellite products). A positive $SWCRE_{TOA}$ bias indicates that the ACCESS-AM2 model is allowing too much shortwave radiation to pass through the clouds and not reflecting enough shortwave radiation out to space. This corresponds to too much shortwave radiation reaching the surface. We have excluded any land regions as significant cloud and radiative biases were found due to non-marine features such as orography. We only consider the summer period for this paper due to the much larger biases found in this season (see F22). Our analysis has been limited to the region of 30-69°S.

The cloud fields of interest include the grid box mean liquid and ice cloud fractions (CFL and CFI), liquid and ice cloud optical depths (TauL and TauI), and cloud top pressure (CTP). These are described in detail in F22, including the pre-processing performed and the decision making around what specific data set to use. As described in F22, the model's COSP liquid water path (LWP) and ice water path (IWP) showed considerable biases when compared to the observed COSP products. This is thought to be a continuation of poor retrievals of the cloud effective radius. While we acknowledge that this bring uncertainty into our results, we have greater confidence in the raw model fields in this instance. For this reason the raw model output was used for these fields (LWP and IWP) instead.

In addition to the cloud fields described above, we use the cloud top pressure - cloud optical depth histogram derived cloud types described in F22. These cloud types were developed using $k$-means clustering, where 12 cloud types were found using the MODIS data set. The 12 cluster centers defined by k-means were then applied to the respective ACCESS-AM2 product, so that each data point was assigned the cluster (aka cloud type) that most closely fit. After initial analysis the 12 cloud types were merged into 10 cloud types. A full description of how these cloud types were found and an analysis of their patterns and relationships to the cloud-radiation bias can be found in F22.

## 2.2 XGBoost

XGBoost, or eXtreme Gradient Boosting, is a highly efficient, fast and scalable algorithm that can handle a large variety of problems (Chen and Guestrin, 2016). XGBoost uses decision trees to predict either categorical or quantitative data, eg. classification or regression problems. Instead of randomly bootstrapping data over many decisions trees to minimise variance, as in the random forest technique, boosting takes a staged approach where each decision tree learns from the mistakes of the previous decision trees to minimise errors, while at the same time boosting higher performing trees (Hastie et al., 2009). Gradient boosting, instead of minimising absolute or squared-errors, uses a gradient descent algorithm to minimise the errors of previous trees (Hastie et al., 2009). XGBoost, an 'extreme' gradient boosting method, is computationally optimised, reduces the amount of data being considered via tree pruning (i.e. removing parts of the trees that were not useful), or the number of nodes in the trees and can reduce the risk of over-fitting (Chen and Guestrin, 2016).

In this work, we use XGBoost to predict the SWCRE$_{TOA}$ biases using the biases in the cloud properties described in the previous section. We refer to these cloud properties as 'features', in line with the language used in machine learning.

To make our prediction, the dataset was split into training and testing datasets, where it was trained on four years of data and tested on one. We have tested the XGBoost model on each of the full summers available at daily resolution: 2015-2016, 2016-2017, 2017-2018 and 2018-2019 (where training was performed on the remaining years) to avoid over-fitting.

Model tuning was performed to improve accuracy and efficiency of the XGBoost model. We have used 4-fold cross validation which splits the training data into a further four individual data sets, in effect generating an ensemble. We note that the 'folds' did not split the data at random, but rather into continuous sections in time, so to avoid the risk of over fitting due to auto correlation. We have employed both the SciKit-Learn GridSearchCV function (Pedregosa et al., 2011) and the XGBoost cross validation function (Chen and Guestrin, 2016) to identify the best combination of hyperparameters for our application. We note that the cross validation described here is in addition to testing on different summers. The workflow used for this tuning (and the exact values used for this work) can be found in the available code linked to this study.

Tuning increased the XGBoost model root mean square error (RMSE) by only 0.44 and the explained variance by 0.014%. While this is a small improvement, we recognise that greater XGBoost model improvement could be found by adding more features. We tested this by including the actual MODIS cloud features (ie. not the biases) as predictors, which also resulted in small improvements. However, by adding more features, physical interpretation of the results becomes more difficult. For this reason, a decision was made to reduce the complexity of the XGBoost model (by only using the biases as features) for the benefit of our understanding.

For the following methods and analysis, we have run the tuned XGBoost model over the entire data set. While this may lead to some over fitting (up to 3% of explained variance), we felt it was important to capture some year-to-year variability, as opposed to just one summers worth of data.

## 2.3 SHAP Feature Importance

With our predictive XGBoost model, we can now begin to understand what cloud features are most important in driving the radiative bias. SHAP aims to understand what contribution each feature has made, i.e. how important they are, to the prediction of $x$ at any particular point in time or space (known as a local prediction) (Lundberg and Lee, 2017). For the SHAP analysis, two key outputs need to be considered to understand the results. The first one is a singular 'base' value for the entire data set of the target variable. This is equal to the mean prediction, in this case provided by the XGBoost model. Secondly, for each point in time and space, and for each predictor variable, a SHAP value is given, with the same units as that of the base value. A SHAP value quantifies how important a feature is to the total prediction of that point $x$. It is essential to note that the sum of all SHAP values for a point $x$ do not equal the prediction of $x$, the base value must also be added. In this sense, we can consider local SHAP values to represent information of the prediction away from the mean (the base value), giving us an indication of how they are contributing to the overall variance. In Section 4, we will provide further description of how to interpret the SHAP values, using the results from this study.

An important strength of SHAP analysis is that the resulting SHAP values are additive with respect to feature attribution, making comparisons across features easy to interpret, even if the features themselves have different units. Similarly, averaging across time or space allows for an in-depth analysis of the results. SHAP analysis further includes the ability to explore the feature dependence and feature interaction, again making interpreting complex models more 'humanly understandable'. Of specific interest to this work are the inbuilt functions to cluster SHAP values, allowing us to determine if two predictors are providing the same information to the XGBoost model and the SHAP interaction values. Clustering the SHAP values provides an indication of how independent each predictor is.

SHAP interaction values offer a further insight into what the 'main' contribution from each feature compared to the value of its interactions with other individual features (Lundberg et al., 2020). It is calculated in a similar way to SHAP values, but allocate credit not just to individual features, as normal SHAP values do, but to all possible pairs of features. It produces a matrix for each individual prediction whereby the 'main' (i.e. non-interacting) contribution is represented by the diagonal values, while the interaction value ($\Phi$) is split evenly between each feature (e.g. $\Phi_{LWP,IWP} = \Phi_{IWP,LWP}$), which are represented in the off-diagonal values of the matrix, so that the total interaction value is $\Phi_{LWP,IWP} + \Phi_{IWP,LWP}$ (Lundberg et al., 2020). Summing the SHAP interaction values along a particular feature gives the same value as the SHAP value for that feature for any one point. Summing the entire interaction matrix (including the interaction values and the main values) will give the same value as the sum of the SHAP values. In that sense, it can be useful to think of the 'main' values from the interaction matrix as the SHAP values minus the interaction values.

However, despite the increased ability to interpret how the features are contributing to the prediction, like many statistical methods, strong feature importance does not provide a causal relationship. Physical understanding of the underlying data must also be considered.

## 2.4 Tools and analysis methods

To produce the results presented in this work, we have applied the XGBoost model and SHAP analysis to the entire data set (2015-2019). For both the bias prediction and feature importance analyses performed in this work we have used the python packages for the Dask (Dask Development Team, 2016) scheduling software, in conjunction with Xarray, SciKit-Learn and other packages specifically mentioned below (Hoyer and Hamman, 2017; Pedregosa et al., 2011). We have run this analysis on the National Computational Infrastructure supercomputer Gadi, using 16 CPUs and 44 GB of memory available via their Open On Demand interface. The workflow for this paper can be found at with our exact methods.

We have defined three regions in this work, following F22, with the boundaries of 30-43°S for the mid-latitudes of the the SO, 43-58°S for the sub-polar region, and a polar region of 58-69°S.

In order to fully evaluate the added value of using a powerful, yet complex and computationally expensive model such as XGBoost, we also used a simple multiple linear regression and Pearson correlation to linearly predict and understand the relationships of the SWCRE$_{TOA}$ bias to the COSP cloud biases (MLR, available from Scikit-Learn, Pedregosa et al., 2011).

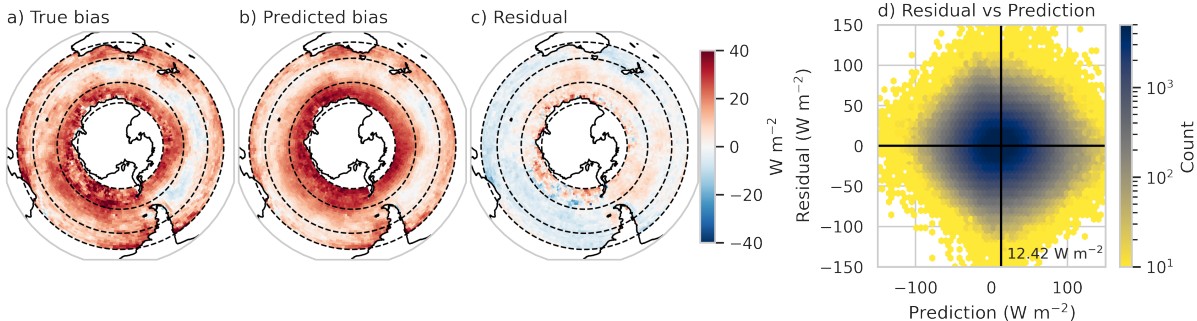

**Figure 1.** The true (a) and XGBoost predicted (b) DJF SWCRE$_{TOA}$ bias (CERES-Syn1D minus ACCESS-AM2) averaged over time; c) shows the residual difference between the predicted and true biases. The dashed lines represent the three regions of interest, mid-latitudes (30-43°S), sub-polar (43-58°S) and the polar (58-69°S) regions. In (d) a histogram of the residual against the prediction, the black lines are represent 0 W m$^{-2}$ for the residual and the mean prediction of 12.42 W m$^{-2}$. All units are in W m$^{-2}$.

## 3  Predicting the SWCRE$_{TOA}$ biases

Figure 1a shows the difference in DJF SWCRE$_{TOA}$ between the ACCESS-AM2 and satellite product (CERES-Syn1D). As a reminder, the SWCRE$_{TOA}$ is calculated as the clear-sky radiation minus the all-sky radiation fields, which results in negative SWCRE$_{TOA}$ values (see Figure A1). The biases are calculated as model minus observations, with positive SWCRE$_{TOA}$ biases indicating that the model has less negative values than the observations, resulting in a positive SWCRE$_{TOA}$ bias, indicating that less sunlight is being refelted out to space.

As discussed in F22 (Fiddes et al., 2022), a strong bias in the polar region of the SO is found, corresponding to too little shortwave solar radiation being reflected back out to space by clouds, and too much being absorbed into the Earth system, including reaching the surface of the ocean. In the sub-polar region, a zonally asymmetric bias is found with positive biases shown in the eastern Indian Ocean and Pacific Ocean sectors, while negative biases are found throughout the rest of that region, as well as in the mid-latitude region. Examination of the model and satellite fields separately (see Appendix Figure A1) shows that the asymmetrical bias appears to be due to ACCESS-AM2 failing to capture the observed spatial variability. We will consider causes of this asymmetry again in Section 4.2.

The spatial variability of this bias suggests that cloud/radiative regimes strongly vary across the Southern Ocean. The asymmetry of this bias makes it difficult to evaluate and understand without splitting the region up into specific synoptic or cloud regimes, as done in studies such as Bodas-Salcedo et al. (2016) or Fiddes et al. (2022). Building on knowledge from previous work, where we understand that particular cloud characteristics (or lack of) are responsible for driving the SWCRE$_{TOA}$ bias in parts of the Southern Ocean, in this study we use biases in such cloud characteristics to predict the SWCRE$_{TOA}$ bias. We hypothesise that if we can satisfactorily predict a bias, we can then use the derived XGBoost model to better understand the sources contributing to the bias. Firstly however, we want to understand more about the individual relationships of each cloud feature bias and the SWCRE$_{TOA}$ bias.

Correlations have been calculated for each cloud feature (shown in Figure A3) with respect to the $SWCRE_{TOA}$ bias, as well as to each other to test for both linearity of the cloud-radiative bias relationship and collinearities between the cloud features. We find that the LWP bias has the strongest relationship with the $SWCRE_{TOA}$ bias, but can only linearly explain 30% of the variance. The other cloud features explain very little of the $SWCRE_{TOA}$ bias under a linear assumption. Similarly, correlations between cloud features do not exceed 31%, indicating weak collinearity. Analysis of the mean $SWCRE_{TOA}$ bias versus the cloud feature biases for each of the cloud types developed by F22 over three latitudinal areas is shown in Appendix Figure A2. This figure confirms, even when only considering the means across cloud types/locations, that these relationships are in some cases highly non-linear, and in other cases, very weak.

In reality, how clouds interact with radiation is a complex system that depends on many, not individual, variables. Hence, analysis comparing singular features should not be expected to provide strong indications of how this system works. For this reason, we also wanted to test whether a simple technique such as multiple linear regression (MLR) could predict the $SWCRE_{TOA}$ bias more satisfactorily. The benefit of such a technique is the low computational requirements and easy initial interpretation. The MLR was able to predict between 42-43% of the variance (when tested on different summers, in the same way as described for the XGBoost training and testing data sets). We have provided more detail on how the MLR prediction performs in the supplementary material. The improvement of the MLR compared to a linear prediction from the individual cloud features does give us an expectation we can improve upon this problem with a more sophisticated tool that can account for the inherent characteristics of the data.

The tool we have chosen to address these issues is XGBoost because it can handle non-linear applications and its performance isn't significantly impacted by collinearity among predictor variables, although as we discuss shortly, these must be still considered when we interpret the importance of these predictors. Using XGBoost, we model the DJF $SWCRE_{TOA}$ biases using the biases in the cloud features discussed above. The results for each period of training and testing were similar, predicting between 54-55% of the $SWCRE_{TOA}$ bias (the $R^2$) and a root mean squared error of between 29.45-30.12. For the subsequent results and analysis, we use the full data set, where 58% of the $SWCRE_{TOA}$ bias is explained.

Figure 1 shows the XGBoost predicted bias (b), the residual between the true and predicted bias (c) and a histogram of the residual versus the predicted bias (d). If we were to consider just the median values of the entire region, the XGBoost model predicts a $SWCRE_{TOA}$ bias of $11.4\,W\,m^{-2}$ compared to $11.7\,W\,m^{-2}$ (the means are $12.4\,W\,m^{-2}$ and $12.4\,W\,m^{-2}$ respectively), implying that the XGBoost model performs quite well, albeit with a lower standard deviation ($32.9\,W\,m^{-2}$) than that of the true bias values ($44.4\,W\,m^{-2}$). The area weighted statistics for the entire region for the true and XGBoost predicted values respectively are: means of $12.0\,W\,m^{-2}$ and $11.5\,W\,m^{-2}$; medians of $11.1\,W\,m^{-2}$ and $10.6\,W\,m^{-2}$ and standard deviations of $45.2\,W\,m^{-2}$ and $33.5\,W\,m^{-2}$. In Figure 1d, a more symmetrical concentration of residuals (y-axis), centred around zero, and a narrow range of predictions is an indicator of a well performing model (x-axis). We can see that the XGBoost model does provide a relatively symmetric pattern, with little skew in any direction. This is especially the case when compared to the MLR, shown in the supplementary material.

We get a much clearer picture of the XGBoost model's performance in different spatial regions in Figure 1a, b and c. Here we can see that the XGBoost model appears to capture the SO negative bias in the polar region reasonably well, with the

predicted median of $23.2\,\mathrm{W\,m^{-2}}$ and a true value of $19.4\,\mathrm{W\,m^{-2}}$. The residual is fairly uniform except in the Weddell Sea region, which could reflect the influence of sea ice. In the sub-polar region, there are small differences between the predicted and true $\mathrm{SWCRE}_{TOA}$ biases, with a median predicted bias of $6.9\,\mathrm{W\,m^{-2}}$ compared to the true bias of $7.3\,\mathrm{W\,m^{-2}}$. Interestingly, these small residuals are not zonally symmetrical, with positive values in the Australian and Pacific Sectors and negative biases in the Atlantic and most of the Indian Ocean sectors. In the mid-latitude region, the positive $\mathrm{SWCRE}_{TOA}$ bias is slightly underestimated in the XGBoost model (median of $8.6\,\mathrm{W\,m^{-2}}$ for the predicted bias compared to $12.6\,\mathrm{W\,m^{-2}}$). The residuals, however, are more zonally symmetric than the sub-polar and polar latitude bands.

As stated previously, our XGBoost model can explain just over half of the variance in the $\mathrm{SWCRE}_{TOA}$ bias ($\mathrm{R}^2\ \tilde{0}.55$). The remainder of this variance, as well as some of the spatial differences observed in Figure 1c suggests that the biases in the cloud fields used in this work may not be the only contributors towards this particular feature of the radiative bias. Alternatively, it could also imply that the robustness of the data that we consider truth are not the same for all regions. The study that the current work follows on from, F22, showed little zonal asymmetry in the LWP, IWP, CFL and CFI biases and similarly, little asymmetry in the relative frequency of occurrence biases of different cloud types, suggesting the need to account for additional physical properties and relationships in these regions.

The missing component could be a range of things, for example, another microphysical cloud field, dynamical field or environmental factor, i.e: the presence of ice nucleating particles, sea surface temperatures or thermodynamical properties of the atmosphere. For example, the influence of dust derived ice nucleating particles may be greater in particular sectors of the SO, in certain seasons, which may change cloud properties in those regions. Alternatively, Zipfel et al. (2022), in predicting LWP over marine boundary layer clouds in the Southeast Atlantic, found that dynamical characteristics and sea surface temperature were important environmental factors, along with other cloud property predictors.

Although the XGBoost model is able to explain a larger amount of the $\mathrm{SWCRE}_{TOA}$ bias than any individual or linearly associated cloud feature could, we must keep two things in mind. Firstly, by adding more (and in particular, not internally consistent) data we make interpretation more difficult and secondly, that the relationships shown here do not prove a causal relationship. We must understand what the model is inferring physically for us to make such association. In this work, we show that even with only 54-55% of the variance explained by these cloud biases we can derive valuable information in how they contribute or do not contribute to the radiative bias. For these reasons, we will continue with our described method, and explore how each of these cloud features contribute to predicted $\mathrm{SWCRE}_{TOA}$ bias using SHAP analysis.

## 4  Understanding the $\mathrm{SWCRE}_{TOA}$ biases

SHAP feature importance analysis allows us to quantify the contribution (or importance) of each cloud feature made to the predicted result, in this case the XGBoost radiative bias. This is demonstrated in Figure 2a, where a so-called force plot is shown. Here, we can see how, for this singular prediction of $x$, each cloud feature has contributed to the total prediction $f(x)$, by shifting the values away from the base value ($12.4\,\mathrm{W\,m^{-2}}$). We note that for the cloud features, their actual values, not the SHAP values are shown in text for the top predictors, while the magnitude and sign of the SHAP value is indicated by the bar.

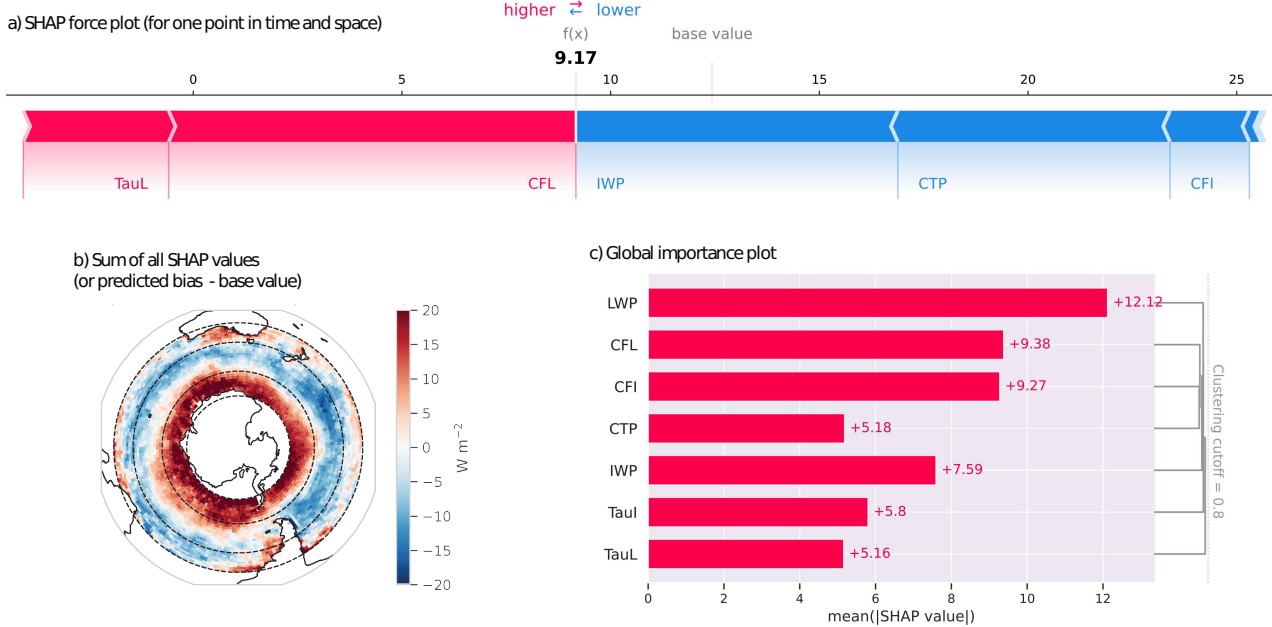

**Figure 2.** (a) a SHAP force plot showing the SHAP values for each predictor for a single example prediction. The sum of these values is the difference between the 'base' value (i.e. mean prediction) of $12.4\,\mathrm{W\,m^{-2}}$ and the individual prediction of $9.17\,\mathrm{W\,m^{-2}}$. (b) the sum of all SHAP values for each spatial point. (c) the global importance plot showing the mean of the absolute SHAP values (in $\mathrm{W\,m^{-2}}$) for each predictor across all predictions, with the dendrogram indicating the degree to which these predictors are clustered.

The base value can be considered as the starting point of any prediction, i.e. if we had no information about the cloud state, a good prediction to start with is the base value. Each cloud feature then adds subsequent information to the prediction, which all together sum (with the base value) to the total prediction. This also means that when summed together for any individual point, or subsection of points, the SHAP values do not represent the total prediction for that point or subsection, but rather the difference from the base value. This is demonstrated by Figure 2b, which shows the mean sum of all SHAP values spatially,
which is equivalent to the difference between the predicted bias (shown in Figure 1b) minus the base value. This characteristic of the SHAP analysis must be kept in mind when considering the SHAP values.

## 4.1 The mean importance of cloud features to the $\mathrm{SWCRE}_{TOA}$ bias

With this functionality of SHAP features in mind, we can now start to analyse our results both globally (ie. the mean across all points) and locally (using subsections, either spatially, temporally or other groupings). Figure 2c shows the global importance
values, which is simply the absolute mean (|M|) of all SHAP values for each cloud feature. These values give us the first indication of how important each cloud feature is comparatively, with the higher the |M| value, the more important that feature

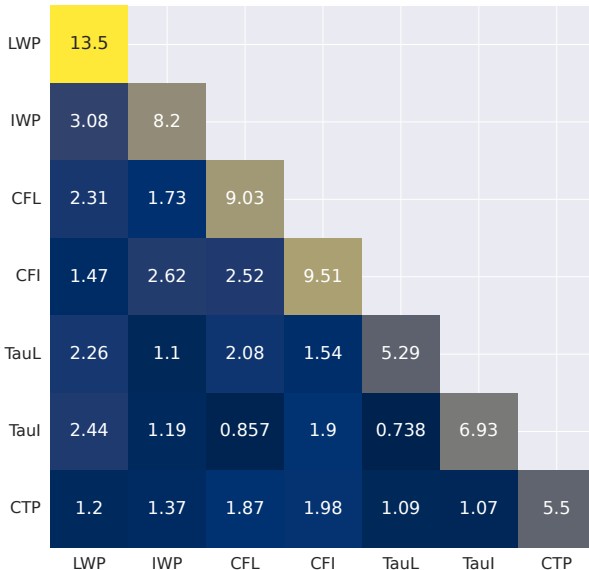

**Figure 3.** A heat map of the absolute mean values for the SHAP interaction matrix. The diagonal values represent the 'main' contribution, i.e. the contribution to the total SHAP values that can be attributed to that individual feature. The off diagonal values represent total interaction value for any two features combined.

is to the overall prediction. Here we can clearly see that the biases in liquid water path contribute most to the predicted $\text{SWCRE}_{TOA}$ bias, with |M| equal to $12.12 \, \text{W m}^{-2}$. This is followed by the liquid cloud fraction ($9.38 \, \text{W m}^{-2}$), ice cloud fraction ($9.27 \, \text{W m}^{-2}$) and ice water path ($7.59 \, \text{W m}^{-2}$), ice optical depth ($5.80 \, \text{W m}^{-2}$), cloud top pressure ($5.18 \, \text{W m}^{-2}$) and liquid optical depth ($5.16 \, \text{W m}^{-2}$) biases.

Figure 2c has not been arranged in order of most to least important, but has instead been clustered, using the in-built SHAP clustering function. This is shown by the dendrogram on the right of the plot, where features are hierarchically merged into clusters. Clustering the SHAP features together can give us an indication of which cloud features are providing similar information to our XGBoost model. If two features provide the same information, the XGBoost model will only use one of them for efficiency, which, while maintaining statistical robustness (eg. avoiding the effects of collinearity), can impact our interpretation of the results. For this plot, we have used a clustering distance cut-off of 0.8, allowing us to see that our features merge at a distance closer to one. A distance of one would imply complete feature independence, while zero would imply complete redundancy. Here we see that one of the least important cloud feature (CTP) is the first to be merged into clusters with the most important cloud features (eg. CFI). However, the merge is occurring only slightly before other features are merged into the other clusters, indicating that even the weakest cloud features are providing independent, if not as important, information to the XGBoost model.

We can further investigate the nature of feature interaction by using the SHAP interaction values. SHAP interaction values are similar to SHAP values, but provide the 'main' contribute from each feature (along the diagonal) plus the interaction values

for each combination of features, for every point in the data set. How to interpret these values is explained in more detail in Section 2.3. In Figure 3 we show the absolute mean values across the whole data set. We note that the off-diagonal values show the total interaction between the two features (see Section 2.3 for more details on how this is calculated).

Figure 3 shows us that the LWP has the highest 'main' interaction, indicating that it provides the most important, individual contribution overall. This is then followed by the CFI, CFL, IWP, TauI, CTP and TauL, which is a slightly different order to that provided by the overall SHAP values in Figure 2c, though this is only occurring when SHAP values are very close in magnitude. Importantly, we can see that the absolute mean of the interaction values are overall, smaller than the main values, indicating that feature interaction is less important to the overall result than the main contributions. If we consider the off-diagonal values in Figure 3, we can see that the largest value is between LWP and IWP, though this value is still smaller than even the lowest 'main' SHAP value. We believe this analysis supports our earlier finding of only weak-moderate collinearity between cloud features and means that we are able to make inferences about how each cloud feature is impacting the radiative bias physically.

## 4.2 Drawing physical conclusions from our SHAP values

While these global importance values are useful to summarise results, they limit the information needed to be able to physically explain why a particular feature may or may not be important. A strength of the SHAP analysis is that an individual SHAP value can be calculated for each feature for every point in time and space. Thus, we can use them to understand how these cloud features contribute to the radiative bias spatially (i.e. their 'local' values), and use the relationships of the SHAP values to their respective cloud feature to try to understand their contribution physically. Figure 4 summarises the outcome of the SHAP analysis.

The density plots of Figure 4 (bottom row) help us understand the spatial means that are presented along the top two rows. Here we find that most of the relationships shown are non-linear, especially with respect to the outliers, with at least one case being parabolic (CTP in Figure 4u). Interestingly, the ice and liquid counterparts for each field (bar cloud top pressure), are similar in shape, giving us confidence that this method is able to capture the role cloud phase plays in the SWCRE$_{TOA}$ bias.

Examining Figure 4 field by field, for LWP, we can see that the spatial SHAP values and the bias patterns line up closely (plots a and h), indicating that increasingly large and positive values of the LWP bias do contribute to an increasingly negative radiative bias (and vice versa). Figure 4o indicates that this behaviour should be expected, with the distribution diagonally centred on zero. In the mid-latitudes, positive LWP biases are associated with negative SHAP values, which transition to negative LWP biases and positive SHAP values in the polar region. Both of these outcomes make sense with our physical knowledge of how LWP interacts with radiation, where clouds with high LWP would increase the amount of sunlight being reflected out to space.

For the CFL, SHAP values are negative in the mid-latitudes while positive in the sub-polar and polar regions. The change in sign of the biases in CFL does not coincide spatially with the SHAP values, which are strongly negative in the polar and sub-polar region, and weakly negative in the mid-latitudes. Examining Figure 4p shows that the relationship between the biases and SHAP values is offset from zero so that weakly negative CFL biases can produce negative SHAP values. These weakly

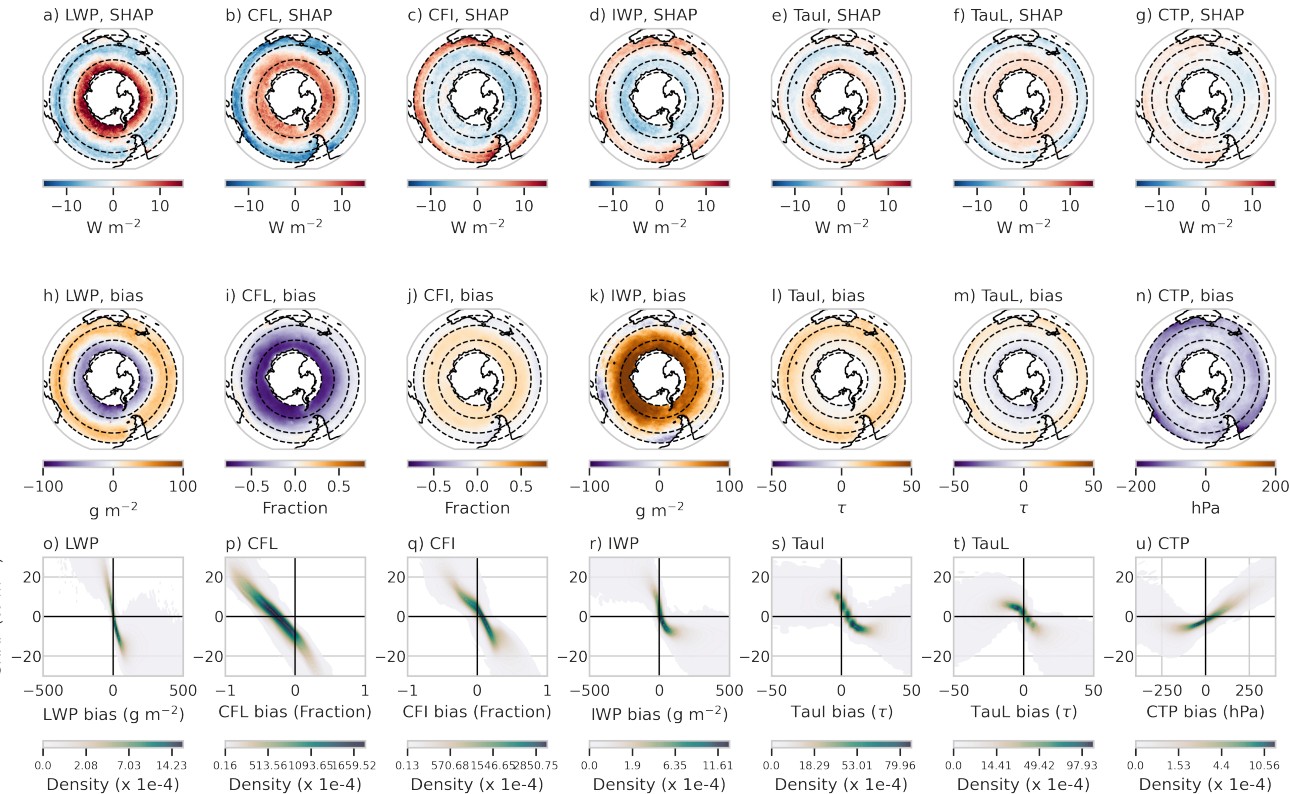

**Figure 4.** Top row (a-g): the time-averaged SHAP values for each cloud feature over the SO domain. Middle row (h-n): the time-mean biases (ACCESS-AM2 - MODIS) for each of the cloud features. Bottom row (o-u): Scatter plot distributions (darker colours indicate greater density) showing the relationship between the bias in each cloud feature and respective the SHAP values. Note that the scatter plots have been limited along the x and y axis to better show the behaviour of the majority of the distribution.

negative CFL biases are occurring predominantly in the northern boundary of the sub-polar region. If we consider making a prediction of the radiative bias, starting from our base value (12.4 W m$^{-2}$), and only knowing about the CFL properties, we can say that in instances where the CFL bias is weakly negative, the radiative bias will be less positive than if the CFL bias is much larger. While this makes sense when considered together with the base value, this result still demonstrates that the radiative influence of clouds is not as simple as "less cloud (even marginally) equals more sunlight passing through", but highlights the ability of the SHAP analysis to capture non-linear processes.

Unlike the CFL, the CFI SHAP values and bias patterns are easier to interpret, with the spatial patterns of Figure 4c and j lining up neatly. Weakly negative biases in CFI correspond to moderately strong positive SHAP values in the mid-latitudes, while positive CFI biases contribute to moderately negative SHAP values. Similarly, the IWP SHAP values are negative in the polar and sub-polar regions and positive in the mid-latitudes. The IWP biases are strongly positive in the polar region, positive

in the sub-polar region and are a mixture of positive and negative values for the mid-latitude, depending on proximity to land masses.

Negative SHAP values are associated with positive IWP and CFI values where too much ice is resulting in too much $SWCRE_{TOA}$ being reflected out to space. The broad similarity in SHAP patterns between the LWP and CFL, as well as the IWP and CFI implies that the underestimation of liquid clouds is a key driver of the SWCRE bias, while the underestimation of ice clouds is actually having a compensating effect. These results suggest that modelling efforts to simply shift water mass from the liquid to the ice phase (eg. by changing ice nucleation temperatures or slowing crystal growth rates) may not entirely

solve the problems in radiative properties. Instead we suggest that these two phase types may need to be tackled independently, with consideration of the ice nucleating particle availability explicitly included in future model development.

    The TauI SHAP values presents an almost cubic function compared to the TauI bias in Figure 4s, which is, on average, positive. The weaker the positive TauI bias (ie. the thinner the cloud), the more positive the SHAP values (less sunlight being reflected out to space). Interestingly, the weakest TauI biases occur in the polar region (Figure 4l), despite strongly overesti-

mated IWP (and to some degree CFI). We suggest that the ice water may be dispersed through the overestimated cloud fraction, resulting in lower biases in optical thickness. However, the positive SHAP value (rather than a simply weak negative value) indicates the non-linearity of this system and possibly a process that we are yet to understand. For TauL, we see a similar relationship, where the thinner the cloud (eg. negative biases), the more sunlight is allowed to pass through the cloud (positive SHAP values). Positive SHAP values are found in the polar and sub-polar regions, while this transitions to negative in the

mid-latitude region (Figure 4f).

    Finally, the CTP presents the only zonally asymmetric cloud SHAP values in the sub-polar and polar region, with negative SHAP values in the Pacific sector and West Antarctic region. The relationship between SHAP values and CTP bias is non-linear (Figure 4n), where positive and negative CTP values can correspond to both positive and negative SHAP values. What is causing the difference in SHAP values is difficult to discern at this broad scale, as meridional differences such as this have

not been previously identified in this study. As discussed earlier (Section 3), the XGBoost model does a poor job of capturing the asymmetry of the $SWCRE_{TOA}$ bias. While the CTP does contribute a small amount, it is, tellingly, the weakest predictor of all the fields examined. We speculate that this field is providing some measure of the cloud type, which is supported by the strongly linear relationship with the cloud types derived previously (with total cloud optical depth and cloud top pressure).

    The results presented in this section have provided us with an overall understanding of what the relationships of the SHAP

values are with the respective biases. In most cases, the relationships presented make physical sense, aligning with our under-standing of how cloud biases may influence radiative biases.

### 4.2.1   Evaluating feature dependence

While our conclusions above make physical sense, the SHAP values presented do not tell us about how individual pairs of features may interact with each other to provide these results. Our linear regressions and clustering analysis has suggested that

each of these features have little dependence on one another, however, our knowledge of the physical world would suggest that some interaction exists. To explore this further we use the SHAP interaction values, which provide a quantitative value

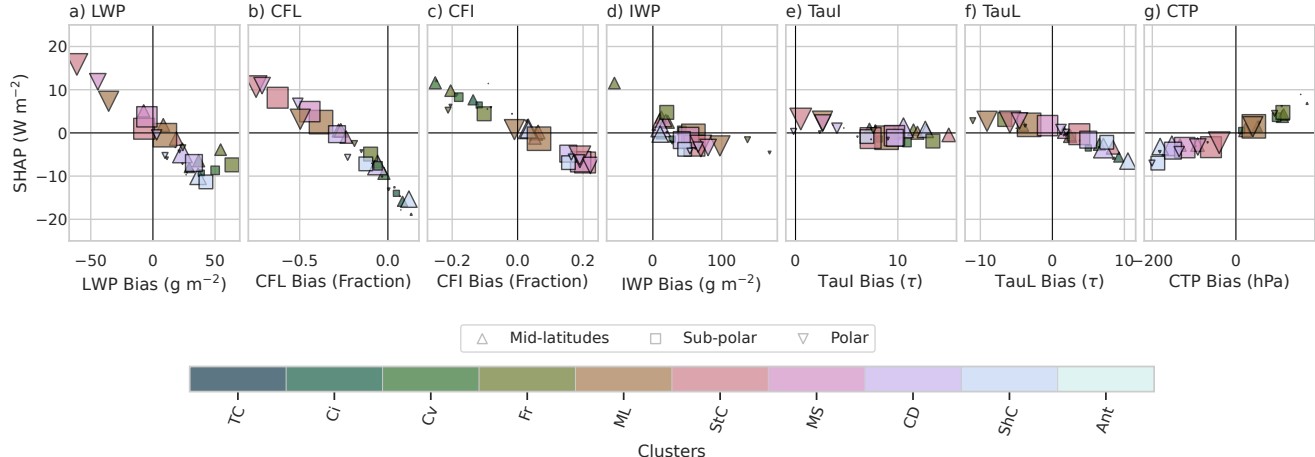

**Figure 5.** The mean SHAP values (y-axis) versus bias (x-axis) for each cloud feature (a-g) averaged by MODIS cloud type (colours) for the three regions of interest indicated by: upwards triangle - mid-latitudes; square - sub-polar; downwards triangle - polar. The size of the shape indicates the frequency of occurrence of each cloud type; a larger shape indicates a more dominant cloud type. The cloud types are as follows: thin cirrus, TC; cirrus, Ci; convective, Cv; frontal, Fr; mid-level, ML; stratocumulus, StC; marine stratiform, MS; cloud decks, CD; shallow cumulus ShC; Antarctic, Ant

of the interaction, as well as the 'main' component from an individual feature alone. As Figure 3 indicated, the main SHAP component for each feature is the dominant driver of the overall SHAP values. We have explored this further by reproducing Figure 4, but for the main component alone. This plot is shown in the Appendix (Figure A4) and shows very little difference to that of Figure 4 in terms of pattern, but does show slightly increased SHAP values. With this result we can be confident that the relationships we are deriving are attributable to the individual feature, and that the interactions between features do not make up the majority of the 'information' provided.

### 4.3 Cloud-scale dependence

In the remaining analysis, we use the F22 MODIS derived cloud types to summarise the SHAP values and biases to determine if a cloud-type dependent relationship exists. Figure 5 shows a scatter plot of SHAP values versus the cloud feature bias, similar to the bottom row of Figure 4. However, in Figure 5, the points represent the mean SHAP versus bias value relationship for each of the MODIS derived cloud types, shown by the colours, with the shapes representing the three regions of interest and sized by their frequency of occurrence. The cloud types are arranged from the highest, thinnest clouds (thin cirrus, TC; cirrus, Ci), to optically thick mid-high clouds (convective, Cv; frontal, Fr), optically thick mid-low level clouds (mid-level, ML; stratocumulus, StC) through to optically thick low level clouds (marine stratiform, MS; cloud decks, CD) and less optically thick low clouds (shallow cumulus, ShC). Finally the Antarctic (Ant) clouds represent mid-level very optically thin clouds. More details about the properties, frequencies and distribution of these clouds can be found in F22.

Figure 5 offers a new, quantitative insight as to how the biases for each cloud type are contributing to the overall SWCRE$_{TOA}$ bias. Firstly, it is clear that there is a dependence on cloud type for the role played by the cloud biases on the radiative bias.

For most cloud features shown, the cloud types are grouped by height/thickness. For example, Figure 5a, the LWP, shows that the mid-level, stratocumulus and marine stratiform clouds in the polar region contribute most to the SWCRE$_{TOA}$ bias, with SHAP values of up to 15.8 W m$^{-2}$ associated with large negative LWP biases. For the sub-polar and mid-latitude regions, lower LWP biases correspond to lower SHAP values for these same cloud types, despite their continued dominance (indicated by size), indicating that for these regions, these optically thick, mid-low level clouds are not driving the SWCRE$_{TOA}$ bias, but

rather the CFL is. The remainder of the cloud types exhibit the opposite trend, where both the higher clouds (eg. convective, frontal) and the lower, less optically thick clouds (cloud decks, shallow cumulus) are characterised by positive LWP biases and negative SHAP values. These cloud types are much more dominant in the sub-polar and mid-latitude regions, indicating a regional dependence with respect to the role that LWP path plays in radiative bias.

The CFL, shown in Figure 5b, shows a similar trend to that of the LWP, where the polar mid-low level, optically thick

clouds demonstrate the largest SHAP values and biases. With this figure, we can also begin to understand the conditions where negative CFL biases result in negative SHAP values, where the optically thinner shallow cumulus, frontal and convective cloud types of the sub-polar region, and interestingly, cloud decks of the mid-latitudes are the major contributors.

For the CFI, Figure 5c, clear groupings of cloud types are found with less dependence on region. Higher clouds, including cirrus, convective and frontal clouds are primarily responsible for driving positive SHAP values, associated with negative

biases. Most other mid-latitude clouds, in addition to the mid-level polar and sub-polar clouds show little bias/SHAP value, while the majority of low-level polar and sub-polar clouds are associated with too much CFI, resulting in negative SHAP values. Interestingly, less clear trends are found for the IWP.

The two optical depth cloud features show a larger regional dependence, where points are grouped more so by region than by cloud type. The SHAP values of all cloud types for the optical depths are small (regardless of bias) compared to the other

cloud features. This may indicate that how the optical depth biases interact with radiation biases is much less dependant on the cloud type than other physical characteristics (eg. latitude).

Finally, the CTP shows the clearest separation of SHAP/bias relationship by cloud type of all the cloud features. The cloud types are clearly grouped together, with the lowest clouds having negative CTP biases/SHAP values and the higher clouds having positive CTP biases/SHAP values. CTP, in earlier analysis, was characterised as the weakest contributor of the radiative

bias, but these results show that if appropriately grouped, it has a similar importance as that of the CFI. This finding indicates that CTP (or cloud vertical structure) may be of greater importance for the radiative bias than currently acknowledged in the literature.

These results offer a new perspective of how different cloud properties, for different clouds and regions may affect the radiative bias. The SHAP analysis allows us to consider many properties at once, in a meaningful and quantitatively comparable

way. Most previous work, including the work preceding this analysis, F22, has been unable to provide such an all-inclusive, and yet quantitative analysis of the cloud-radiative bias to date. For example, while F22 was able to highlight the importance

of stratocumulus and mid-level clouds for the SO cloud-radiative bias, as well as the compensating effects of lower, thinner clouds such as shallow cumulus clouds, the analysis was far more qualitative than what has been presented in this work.

## 5   Using SHAP analysis to understand model perturbations

We have demonstrated the usefulness of the SHAP method in quantitatively understanding drivers of biases. This method has provided useful insights into the mean drivers of the $SWCRE_{TOA}$ bias as well as into cloud-specific influences. We now want to test if this method is useful in understanding how perturbations applied to a model may change the results. To do this, we have performed a second nudged simulation, where we have altered the ice-crystal capacitance, following the work of (Varma et al., 2020). We refer to the new simulation as the 'ice' simulation (compared to the 'control'). In the ice simulation, we

have changed the capacitance from 1 in the control, which assumes a spherical crystal, to 0.5, which assumes an oblate ice crystal. The effect of this change is to slow down ice crystal growth rates. Overall, reducing the ice crystal growth rates reduces the radiative bias by -1.0 W m$^{-2}$ for the entire region (-1.2 W m$^{-2}$ for polar region, -1.0 W m$^{-2}$ for sub-polar region and -0.8 W m$^{-2}$ for mid-latitudes) in our simulations. This difference is smaller on average and noisier than that found in Varma et al. (2020) (and not statistically significant), which we expect is a result of the nudging of the model. Similar constraints on

meteorology was found in Fiddes et al. (2021) when nudging was applied during model perturbation experiments. This study compared perturbed nudged simulations to perturbed free-running simulations. The nudged simulations had a smaller overall response, with a lot of seemingly random variation, which, as in this study, we are referring to as 'noise'. Regardless, this small perturbation experiment provides an opportunity to test the usefulness of the SHAP method, even if the changes are very small.

With this second simulation, we have re-trained (and re-tuned) our XGBoost model. The performance of the XGBoost-

ice model is very comparable to that of the control, explaining between 54-55% of the variance. The mean difference in the predicted bias from the control is similar to that of the true bias, of 0.9 W m$^{-2}$. Once again, LWP is the most important cloud feature, and the order of importance for each cloud feature has not changed.

Figure 6 shows the changes in SHAP values (ice-control) for each cloud field along the top row, and the changes in the fields themselves along the middle row (also ice-control). The noisiness introduced by the nudging is clear in these plots and we note

that none of the changes are statistically significant. Nevertheless, some interesting features can been seen in these results.

Firstly, we can see that changing the ice crystal shape, which slows ice growth and increases LWP in the polar and sub-polar regions (Figure 6h). This change reduces the overall bias in the polar region (which is negative - see Figure 4h). In the mid-latitudes the change is much less coherent. The resulting change in SHAP values indicates that in the polar region, the increased LWP has overall reduced the SHAP values, although results are quite noisy. Figure 6o demonstrates that polar and

sub-polar stratocumulus clouds are associated with the largest increase in LWP and reduction in SHAP values. Interestingly, shallow cumulus clouds, despite showing only small increases in LWP, are also associated with larger SHAP values for all regions.

CFL (Figure 6i) shows a varied and very small response to the change in ice crystal growth. CFI on the other hand (Figure 6j) shows a clear and consistent increase in fraction. This increase in CFI may be caused by an increase in cloud lifetime as

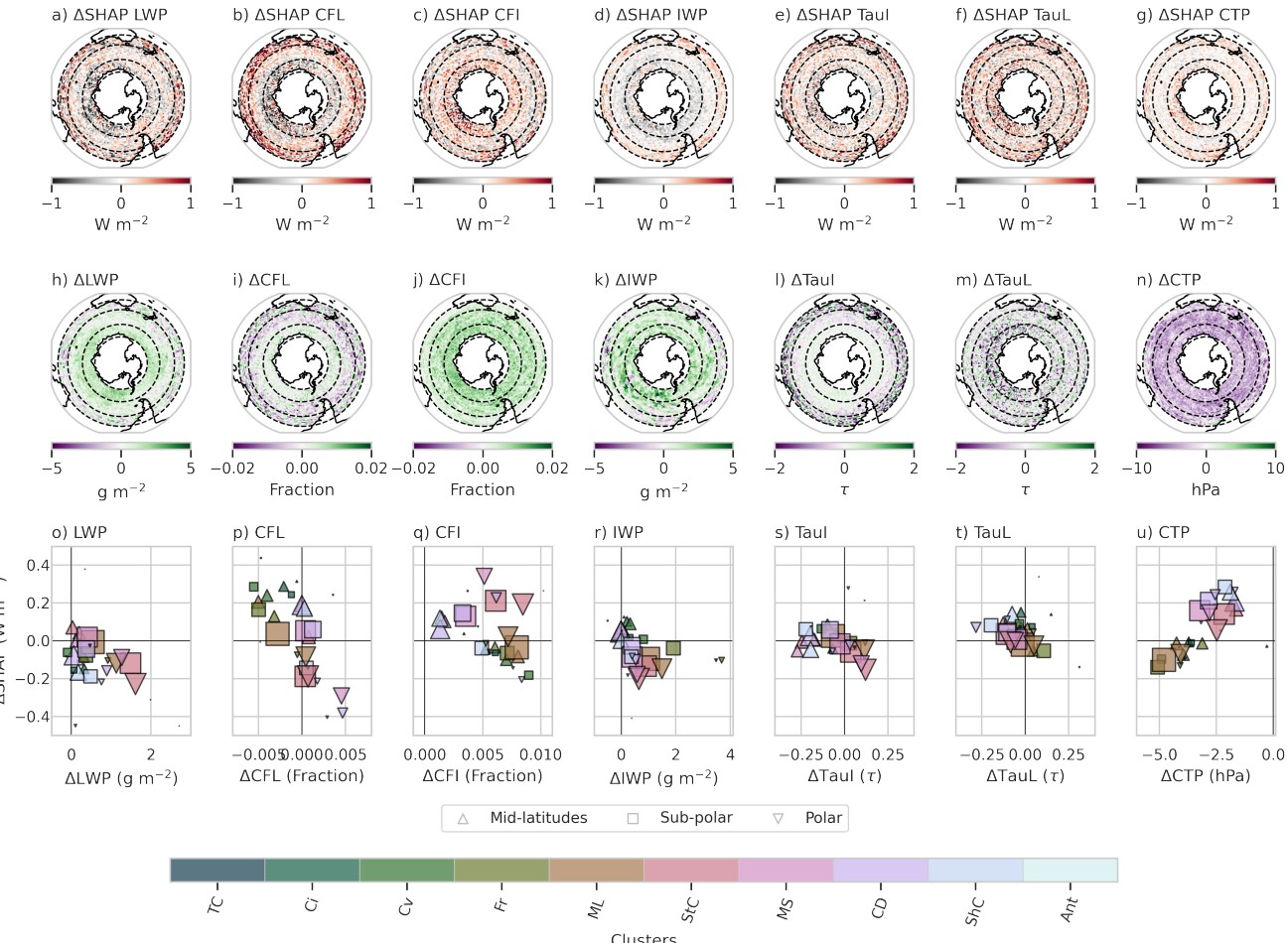

**Figure 6.** Top row (a-g): the time-averaged difference in SHAP values (ice-control) for each cloud feature over the SO domain. Middle row (h-n): the time-mean changes (ice-control) for each of the cloud features

the ice crystal growth rate slows down, which is supported by a small decrease in snowfall rates over the polar region (not shown). The change in SHAP values for CFL is predominantly negative in the polar regions and positive in the sup-polar to mid-latitudes. For the CFI, the changes in SHAP values are harder to differentiate.

Considering these changes grouped by cloud type, some interesting patterns are apparent, though we note the magnitude of these values in cloud fraction are again very small. For CFL, higher clouds (cirrus, convective, frontal) are found to be associated with a reduction in CFL and an increase in SHAP values. In the case of mid-latitude convective clouds we suggest this is caused by the associated increase in LWP converting into an increase in rainfall, overall reducing cloud lifetime. For the remaining cloud types, the picture is less clear with very little change in rainfall. Polar marine stratiform and cloud decks exhibit a stronger positive change in CFL, associated with a decrease in SHAP values, while the remaining cloud types (for

all regions) show a much more neutral response. For CFI, all cloud types, for all regions, experience on average an increase in fraction, but this increase results in positive change in SHAP values for the lower, thicker clouds and a negative change in SHAP values for the higher, thinner clouds. The varying behaviour of higher/thinner clouds compared to lower/thicker clouds for CFI and CFL, both with respect to the cloud properties themselves, and their resulting SHAP values, highlights that a 'one solution fits all' approach to reducing cloud-radiative biases in models is not appropriate.

We will not discuss the changes in optical depth or IWP for brevity, however, will touch on CTP in Figure 6g, n, and u. Consistent reductions in CTP are found for the entire regions, which acts to increase in magnitude the already negative bias (Figure 4n). The changes to SHAP values on average are small. However, if we consider these values separated by cloud type/region, we can see a strong separation in behaviour between cloud types. Thinner, higher clouds, which were on average overestimating CTP, are found to have larger decreases in the CTP. The lower, thicker cloud types exhibit a smaller magnitude decrease, only minimally exacerbating the CTP bias (see Figure 4u). These two groups of clouds types are also associated with different SHAP values, where the lower/thicker clouds, decreases in CTP (increases in cloud height) results in more positive SHAP values. The higher/thinner clouds, despite also decreasing in CTP, are found to have negative changes in SHAP values.

The results presented in this section indicate very small, insignificant changes to cloud properties when ice crystal growth rates are slowed within a nudged model framework. Nevertheless, these changes have resulted in the expected decrease in radiation bias and are of use to us within the context of this analysis: can we use SHAP values to examine changes in model parameterizations? Here, the behaviours of cloud types/regions, with respect to changes in cloud properties/SHAP values, are of greater interest to this work for a number of reasons. Firstly, these results highlight that despite the averages presented in the top two rows of Figure 4, cloud-radiative interactions are much more complex. Importantly, this analysis has been able to condense and analyse a large amount of information in a more succinct way than previously possible, with many prior studies only concentrating on one region and/or one cloud type (typically low-level stratiform). Secondly, this analysis clearly demonstrates instances of compensating radiative errors between different cloud properties, types and regions. While spatial compensating errors have previously been identified, few prior studies have examined them with respect to cloud type. To our knowledge, no other study has been able to directly associate changes in cloud properties to changes in the radiative bias (though we stress that the results here do not present a casual relationship).

We believe that this last point is where the power of SHAP analysis lies. In this work we can see that, for example, for low/thick clouds in the polar region, the resulting changes in LWP, CFL and IWP caused by the model perturbation work to decrease the prediction of the radiative bias, while CFI and CTP have the opposite effect. Our method is able to easily account for individual and interaction, sometimes non-linear, influences from each predictor, while still allowing us to interpret our predictions are made.

## 6   Discussion and Conclusions

The SO radiation bias has been a topic of considerable research over the last decade, motivating a number of innovative methods and studies to understand its controls. While methods such as cloud regime clustering or cyclone tracking have become standard

ways to evaluate the SO radiative bias (eg. as in Bodas-Salcedo et al., 2016), they do not account for the entirety of the SO and the range of biases found across it. This narrower view has proven detrimental to some model development studies, where some aspects of cloud microphysics, such as ice nucleation temperatures, have been altered to target the worst of the radiative
bias, but have led to unwanted changes in other regions (eg. as in Varma et al., 2020; Furtado et al., 2016).

In this study, we have proposed a new method for model bias evaluation, employing modern machine learning and taking advantage of the large amount of cloud and radiation data available to us and facilitated by enhanced computing resources. This method, where we use biases in cloud properties to predict the $SWCRE_{TOA}$ bias, considers the entire SO, from the mid-latitudes to the polar regions, at a daily timescale. This study has been made possible due to our ability to perform a
nudged climate model simulation, an underutilised method in climate research without which coincident in time comparisons to satellite fields are not possible. This work provides a new perspective on the downwelling shortwave radiation bias, which will help guide our future model developments. Furthermore, we expect that this method could be applied to a range of complex model biases throughout the climate system beyond radiation.

We note that the cloud fields used in this work, including the satellite products, and the modelled products each contain in-
herent uncertainties. While the MODIS L3 product has specifically been produced for model evaluation, we must acknowledge that these products may not represent the 'truth'. Greater discussion on this can be found in F22. Similarly, Pei et al. (2023) find an underestimation of short wave cloud radiative effect at the surface of $7.9\,W\,m^2$ in CERES compared to ground observations at Macquarie Island, indicating that similar issues exist in the satellite radiative fields. To add to this, despite satellite simulators such as COSP being designed to reduce the uncertainties between modelled and satellite retrieved products, we have found
that some of these simulated fields were of too poor a quality to be used with confidence. This was the case for the LWP and IWP fields, for which we used the raw modelled products instead. While this decision adds to the uncertainty of our analysis, we are none-the-less confident in our overall results (eg. LWP being a dominant driver of biases radiative biases).

Specifically, in this work we have continued our evaluation of the ACCESS-AM2 models SO cloud and radiative biases (from Fiddes et al., 2022). The ACCESS-AM2 SO $SWCRE_{TOA}$ bias is shown to be the largest in magnitude over the polar
region, while it weakens in the sub-polar and mid-latitude regions. Of interest to this work in particular is that the ACCESS-AM2 model lacks the zonal asymmetry of the CERES-Syn1D $SWCRE_{TOA}$, with positive biases in the Australian and Pacific sectors. This asymmetry has not previously been considered, and is possibly a reflection of the differences in $SWCRE_{TOA}$ biases between cloud regimes, or other unaccounted for physical processes.

Importantly, the XGBoost model suggests that the ACCESS-AM2 SO $SWCRE_{TOA}$ bias cannot be completely explained
by the biases in several key cloud properties, including LWP, IWP, CFL, CFI, TauL, TauI, and CTP. Many of these cloud biases have a non-linear relationship with the radiative bias, as well as weak collinearities amongst them, demonstrating the complexity of the system and the need for a method that can take such complexities into account. For this purpose, we have used a tuned and tested XGBoost model to predict the $SWCRE_{TOA}$ bias, using the biases from these cloud fields as predictors.

The XGBoost model can explain up to 55% of the $SWCRE_{TOA}$ bias; more than any one of the cloud fields alone or a linear
combination could. While the general pattern of the radiative bias is captured, the zonal asymmetry is not, with the XGBoost prediction lacking the positive values found in the sub-polar region. This finding suggests that the asymmetry may not be a

function of the cloud properties explored in this work, but possibly some other environmental factor, such as a dynamical or thermodynamical property, aerosol sources and interaction or proximity to the polar front. Other studies have found that environmental factors, such as SSTs or dynamical predictors are important for predicting cloud characteristics or radiation (Zipfel et al., 2022; Mallet et al., 2023). Such factors may contribute to the missing 45% of predictability, though this has not been tested in this work. For this analysis, we have chosen not to explore additional parameters for three reasons: 1) to maintain interpretability - the more parameters you have the less interpretable your model becomes; 2) to maintain some level of data homogeneity - we try to limit the number of sources (eg. different satellite products) our predictors are coming from to limit inconsistencies in assumptions; and 3) to keep our focus on the cloud-radiative relationship, without compounding it with other external factors. However, we suggest that future work should explore how environmental factors may influence the $SWCRE_{TOA}$ bias, including looking at cloud condensation nuclei availability, cloud droplet number concentration, ice nucleating particle concentration, sea surface temperature, and vertical cloud overlap.

One of the benefits of using the XGBoost model is that it can efficiently be interpreted and analysed with the SHAP feature importance method. Our SHAP analysis has shown that the biases in LWP are the main drivers of the $SWCRE_{TOA}$ bias (with CFL, CFI and IWP following). Further exploration of the SHAP feature importance and the cloud biases indicates that the relationships are still non-linear, but for the most part, we can make physical sense of them. In addition, we have shown that there are cloud-type specific behaviours that can be easily captured using this type of analysis, including that for mid-latitude and sup-polar mid-level, stratocumulus and marine stratiform clouds the CFL has higher SHAP values than the LWP, indicating a greater importance. This method allows us to evaluate cloud-radiative biases in a much more holistic way, compared to isolating just one, often pre-designated, 'important' cloud type. Furthermore, we find distinct cloud-type behaviours in the SHAP/bias relationship, which we we expect can be leveraged in future model development.

We have tested this by exploring a simple model perturbation, previously described in Varma et al. (2020), where we reduce the ice shape capacitance from 1.0 to 0.5, thereby reducing the ice crystal growth rates, increasing LWP and reducing the radiative bias by approximately $1\,\mathrm{W\,m^{-2}}$. We note that (Varma et al., 2020) saw much larger radiative changes in response to this perturbation, which we suggest is due to the free-running nature of their experiments. Although the changes between this test run and our control were small (and insignificant), we believe that our results demonstrate the power of SHAP analysis, where complex changes within a system can be evaluated and their impact quantified.

This finding is particularly relevant to methods used to constrain models to observationally plausible values, such as that done in Regayre et al. (2020, 2023). In these studies, perturbed parameter ensembles (PPEs) have been used to sample distributions of many parameters, after which, observations are used constrain the model to internally consistent and plausible values. These studies, which provide an efficient and comprehensive way to both evaluate and tune model parameters, use huge arrays of data representing complex changes in the model. While alternative methods to determine feature importance have been implemented in these studies, an approach such as the one presented in this work would provide an efficient way to interpret the effects of the parameter tuning. We further note recent developments in SHAP in which multiple targets can be predicted and evaluated, potentially providing a significant advantage for studies using PPEs.

In this work, we find that the SHAP values for opposite cloud phases do not balance out. Total liquid phase values outweigh the total ice phase values. This finding, plus the non-linearities and cloud-type dependencies of the system suggest that changing our cloud parameterisations to simply move mass from one phase to another in order to balance the liquid and ice phases may not remove the SWCRE$_{TOA}$ bias entirely. We suggest that concerted effort is required to improve the individual representation

of each phase, in a more physical way, which can take into consideration different environmental conditions. We suggest that the parameterisations of CCN and INP is a good place to start as, unlike other changes to the model (eg. changing the freezing temperatures, detrainment temperatures, ice crystal shape or growth rates: Varma et al., 2020; Furtado and Field, 2017; Kay et al., 2016), they are derived independently, and do not simply change phase partitioning. Note we are not suggesting that improvements to microphysical representations are futile (in fact the opposite); we do suggest that they may have a lesser

impact, or in some cases, undesirable impacts, if not done with strong physical backing.

Vignon et al. (2021) has shown that empirically forcing a model's INP concentrations can result in significant improvements in super cooled liquid water fraction. Such work has begun for the UM model family: (Vergara-Temprado et al., 2017) have explored the importance of marine organics and dust to INP concentrations in the SO and their subsequent control on cloud reflectivity, but a more concerted effort is needed. Specifically, we need to ensure the chemical and aerosol pathways respon-

sible for both INP and CCN are a) accurate and b) coupled to the cloud scheme satisfactorily (two-way coupling preferred). In ACCESS-AM2, analysis has shown that CCN concentrations are significantly underestimated in the SO (manuscript in preparation), while INP concentrations are not resolved by the aerosol scheme, but rather parameterised without consideration of the compositional environment. This is a key area for development, a process that has been started by Varma et al. (2021), though still requires considerable work with a stronger connection to the aerosol scheme. We hope that future work in this

space can make use of the methods presented in this study, to holistically quantify how their changes to the model affect the cloud-radiative system.

To summarise, we have provided a new method for understanding model biases, using a nudged climate simulation and machine learning. We hope that this method can be applied to other fields to gain new insight into the complexity and the drivers of modelling biases, and how model perturbations may improve or worsen such biases. When considered as a whole, the SO

SWCRE$_{TOA}$ bias is shown to be complex, with large non-linearities, cloud-type and regional dependencies and compensating errors. Our results suggest that the liquid phase of clouds is the most important contributor to the SWCRE$_{TOA}$ bias, more so than the ice phase, and that different higher/thinner clouds often behave in an opposing manner to that of the lower/thicker cloud types, in some cases having compensating effects on the radiative bias. We propose that we need to address the biases in the liquid and ice phase of SO cloud properties individually (i.e. more physically) in order to reduce the SWCRE$_{TOA}$ bias and

we expect that the best way to do this is to ensure the nucleating particles (CCN and INP) are resolved by an aerosol scheme and fully coupled to the microphysics.

*Code and data availability.* Processed model and observational data and the relevant code for this project is available at https://doi.org/10.5281/zenodo.7196622. The ACCESS-AM2 model includes the UM vn 10.6 GA7.1 atmospheric model, which is protected

under intellectual property copyright restrictions. The land surface model CABLE is available by registration from
https://trac.nci.org.au/trac/cable (accessed 13th July 2023). Details of how to get started with the ACCESS model can be found at
https://accessdev.nci.org.au/trac/wiki/GettingConnected (accessed 13th July 2023). The ACCESS-AM2 model code and configuration can
be accessed by existing UM users under the bx400 and cg207 suites and was provided to the editor and reviewers.

CERES data can be downloaded from https://ceres.larc.nasa.gov/data/ (last accessed 25th March 2022). MODIS data can be downloaded
from https://ladsweb.modaps.eosdis.nasa.gov/archive/allData/61/MCD06COSP_D3_MODIS/ (last accessed 25th March 2022).

*Author contributions.* SF completed the model simulations, analysis and writing of this work. MM provided guidance on the machine
learning methods, analysis and writing. MW provided guidance on ACCESS setup. MM, AP, SA, MW and KF contributed to the planning,
ideas and revisions of this paper.

*Competing interests.* The authors have no competing interests to declare

*Acknowledgements.* This project received grant funding from the Australian Government as part of the Antarctic Science Collaboration
Initiative program, under the Australian Antarctic Program Partnership, ASCI000002. Resources and services from the National Computa-
tional Infrastructure (Project jk72), supported by the Australian Government, were used. S.F. acknowledges the ARC Centre of Excellence
for Climate Extremes for their maintenance of virtual environments and code and model support. The authors would like to acknowledge
the teams at NASA CERES, NASA Earth data and the CFMIP project for making the tools and data used in this work publicly available.
The contribution of S.P.A. was supported by the Australian Antarctic Science project 4292. We would also like to acknowledge our two
Reviewers who provided encouraging and insightful comments.

# Appendix A: Additional figures

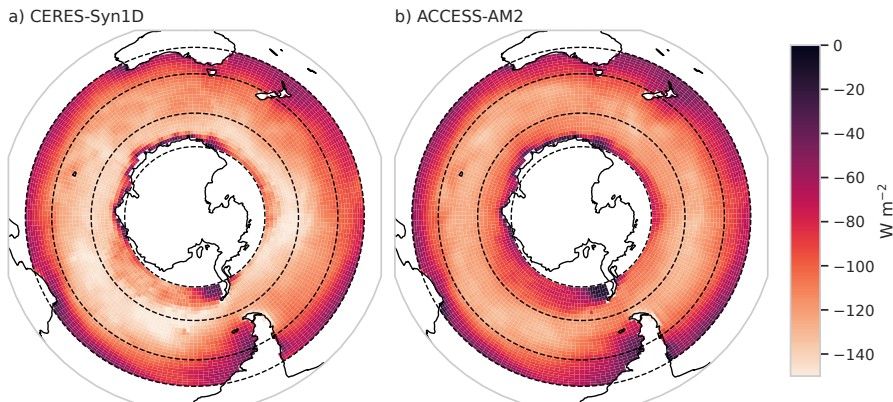

**Figure A1.** The summer time (DJF) ocean SWCRE$_{TOA}$ for the a) CERES-Syn2D satellite product, b) the ACCESS-AM2 model. All units are in W m$^{-2}$

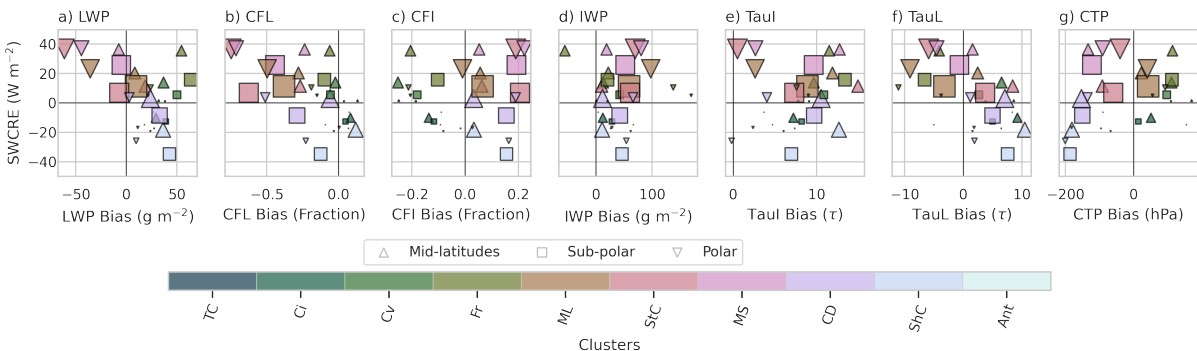

**Figure A2.** The mean SWCRE$_{TOA}$ bias (y-axis) versus the bias (x-axis) for each cloud feature (a-g) averaged by MODIS cloud type (colours) for the three regions of interest indicated by: upwards triangle - mid-latitudes; square - sub-polar; downwards triangle - polar. The size of the shape indicates the frequency of occurrence of each cloud type; a larger shape indicates a more dominant cloud type. The cloud types are as follows: thin cirrus, TC; cirrus, Ci; convective, Cv; frontal, Fr; mid-level, ML; stratocumulus, StC; marine stratiform, MS; cloud decks, CD; shallow cumulus ShC; Antarctic, Ant

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

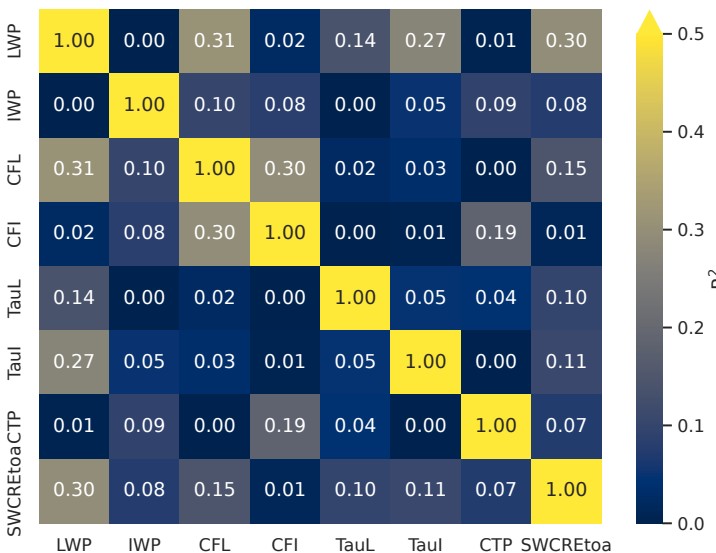

**Figure A3.** Heat maps of the explained variance ($R^2$) between each of the cloud bias features as well as the SWCRE$_{TOA}$ bias

Bodas-Salcedo, A., Webb, M. J., Bony, S., Chepfer, H., Dufresne, J. L., Klein, S. A., Zhang, Y., Marchand, R., Haynes, J. M., Pincus, R., and John, V. O.: COSP: Satellite simulation software for model assessment, Bulletin of the American Meteorological Society, 92, 1023–1043, https://doi.org/10.1175/2011BAMS2856.1, 2011.

Bodas-Salcedo, A., Williams, K. D., Ringer, M. A., Beau, I., Cole, J. N. S., Dufresne, J. L., Koshiro, T., Stevens, B., Wang, Z., and Yokohata, T.: Origins of the solar radiation biases over the Southern Ocean in CFMIP2 models, Journal of Climate, 27, 41–56, https://doi.org/10.1175/JCLI-D-13-00169.1, 2014.

Bodas-Salcedo, A., Hill, P. G., Furtado, K., Williams, K. D., Field, P. R., Manners, J. C., Hyder, P., and Kato, S.: Large contribution of supercooled liquid clouds to the solar radiation budget of the Southern Ocean, Journal of Climate, 29, 4213–4228, https://doi.org/10.1175/JCLI-D-15-0564.1, 2016.

Bodman, R. W., Karoly, D. J., Dix, M. R., Harman, I. N., Srbinovsky, J., Dobrohotoff, P. B., and Mackallah, C.: Evaluation of CMIP6 AMIP climate simulations with the ACCESS-AM2 model, Journal of Southern Hemisphere Earth Systems Science, https://doi.org/10.1071/ES19033, 2020.

Chen, T. and Guestrin, C.: XGBoost: A scalable tree boosting system, in: Proceedings of the ACM SIGKDD International Conference on Knowledge Discovery and Data Mining, vol. 13-17-August-2016, pp. 785–794, Association for Computing Machinery, https://doi.org/10.1145/2939672.2939785, 2016.

Chubb, T. H., Jensen, J. B., Siems, S. T., and Manton, M. J.: In situ observations of supercooled liquid clouds over the Southern Ocean during the HIAPER Pole-to-Pole Observation campaigns, Geophysical Research Letters, 40, 5280–5285, https://doi.org/10.1002/grl.50986, 2013.

Dask Development Team: Dask: Library for dynamic task scheduling, 2016.

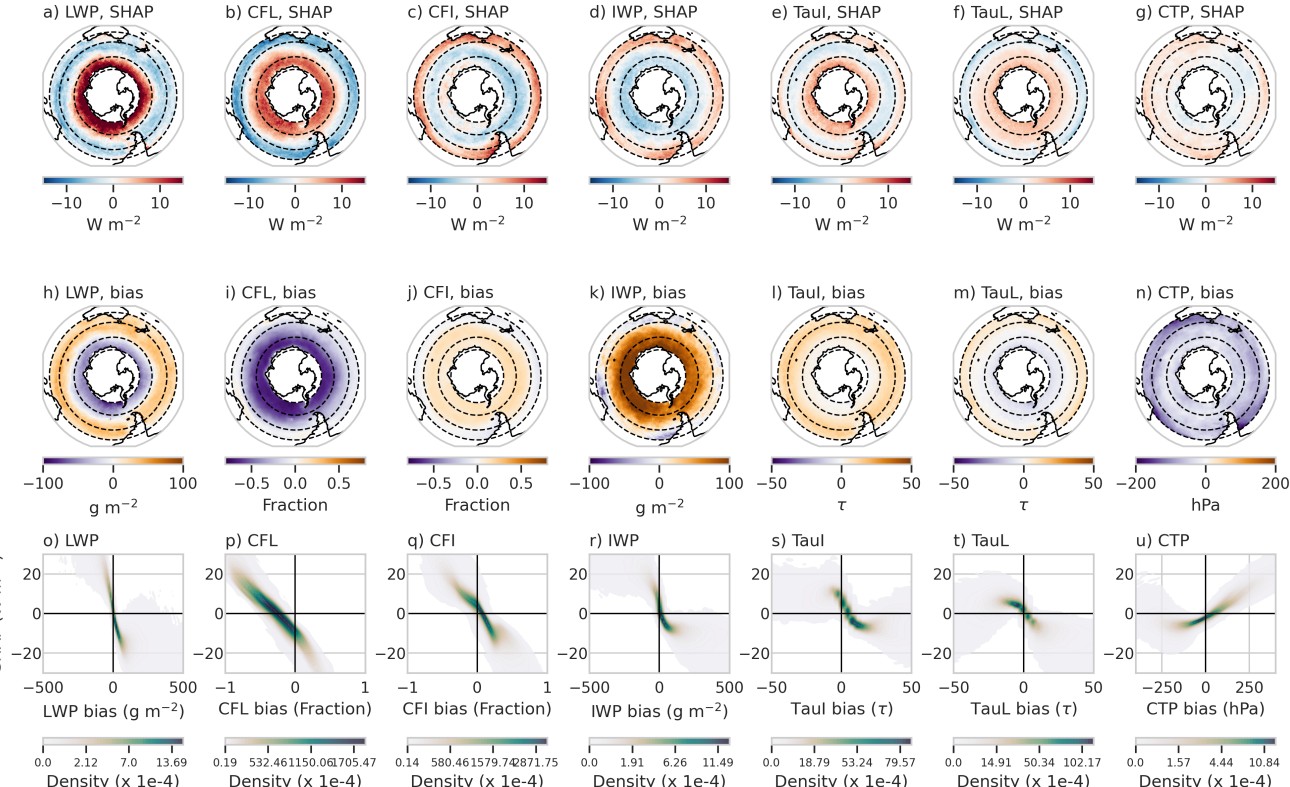

**Figure A4.** Top row (a-g): the time-averaged main SHAP values (as calculated via the interaction analysis) for each cloud feature over the SO domain. Middle row (h-n): the time-mean biases (ACCESS-AM2 - MODIS) for each of the cloud features. Bottom row (o-u): Scatter plot distributions (darker colours indicate greater density) showing the relationship between the bias in each cloud feature and respective the main SHAP values. Note that the scatter plots have been limited along the x and y axis to better show the behaviour of the majority of the distribution.

Doelling, D. R., Loeb, N. G., Keyes, D. F., Nordeen, M. L., Morstad, D., Nguyen, C., Wielicki, B. A., Young, D. F., and Sun, M.: Geostationary enhanced temporal interpolation for ceres flux products, Journal of Atmospheric and Oceanic Technology, 30, 1072–1090, https://doi.org/10.1175/JTECH-D-12-00136.1, 2013.

Doelling, D. R., Sun, M., Nguyen, L. T., Nordeen, M. L., Haney, C. O., Keyes, D. F., and Mlynczak, P. E.: Advances in geostationary-derived longwave fluxes for the CERES synoptic (SYN1deg) product, Journal of Atmospheric and Oceanic Technology, 33, 503–521, https://doi.org/10.1175/JTECH-D-15-0147.1, 2016.

Eyring, V., Bony, S., Meehl, G. A., Senior, C. A., Stevens, B., Stouffer, R. J., and Taylor, K. E.: Overview of the Coupled Model Intercomparison Project Phase 6 (CMIP6) experimental design and organization, Geoscientific Model Development, 9, 1937–1958,
https://doi.org/10.5194/gmd-9-1937-2016, 2016.

Fan, J., Wang, X., Wu, L., Zhou, H., Zhang, F., Yu, X., Lu, X., and Xiang, Y.: Comparison of Support Vector Machine and Extreme Gradient Boosting for predicting daily global solar radiation using temperature and precipitation in humid subtropical climates: A case study in China, Energy Conversion and Management, 164, 102–111, https://doi.org/10.1016/j.enconman.2018.02.087, 2018.

Fiddes, S. L., Woodhouse, M. T., Lane, T. P., and Schofield, R.: Coral-reef-derived dimethyl sulfide and the climatic impact of the loss of coral reefs, Atmospheric Chemistry and Physics, 21, 5883–5903, https://doi.org/10.5194/acp-21-5883-2021, 2021.

Fiddes, S. L., Protat, A., Mallet, M. D., Alexander, S. P., and Woodhouse, M. T.: Southern Ocean cloud and shortwave radiation biases in a nudged climate model simulation: does the model ever get it right?, Atmospheric Chemistry and Physics, 22, 14 603–14 630, https://doi.org/10.5194/acp-22-14603-2022, 2022.

Field, P. R. and Wood, R.: Precipitation and cloud structure in midlatitude cyclones, Journal of Climate, 20, 233–254, https://doi.org/10.1175/JCLI3998.1, 2007.

Fuchs, J., Cermak, J., and Andersen, H.: Building a cloud in the southeast Atlantic: Understanding low-cloud controls based on satellite observations with machine learning, Atmospheric Chemistry and Physics, 18, 16 537–16 552, https://doi.org/10.5194/acp-18-16537-2018, 2018.

Furtado, K. and Field, P.: The Role of Ice Microphysics Parametrizations in Determining the Prevalence of Supercooled Liquid Water in High-Resolution Simulations of a Southern Ocean Midlatitude Cyclone, Journal of the Atmospheric Sciences, 74, 2001–2021, https://doi.org/10.1175/JAS-D-16-0165.1, 2017.

Furtado, K., Field, P. R., Boutle, I. A., Morcrette, C. J., and Wilkinson, J. M.: A Physically Based Subgrid Parameterization for the Production and Maintenance of Mixed-Phase Clouds in a General Circulation Model, Journal of the Atmospheric Sciences, 73, 279–291, https://doi.org/10.1175/JAS-D-15-0021.1, 2016.

Hastie, T., Tibshirani, R., and Friedman, J.: The Elements of Statistical Learning: Data Mining, Inference, and Prediction, Springer Series in Statistics, Springer New York, New York, NY, https://doi.org/10.1007/978-0-387-84858-7, 2009.

Haynes, J. M., Jakob, C., Rossow, W. B., Tselioudis, G., and Brown, J. B.: Major characteristics of Southern Ocean cloud regimes and their effects on the energy budget, Journal of Climate, 24, 5061–5080, https://doi.org/10.1175/2011JCLI4052.1, 2011.

Hersbach, H., Bell, B., Berrisford, P., Hirahara, S., Horányi, A., Muñoz-Sabater, J., Nicolas, J., Peubey, C., Radu, R., Schepers, D., Simmons, A., Soci, C., Abdalla, S., Abellan, X., Balsamo, G., Bechtold, P., Biavati, G., Bidlot, J., Bonavita, M., Chiara, G., Dahlgren, P., Dee, D., Diamantakis, M., Dragani, R., Flemming, J., Forbes, R., Fuentes, M., Geer, A., Haimberger, L., Healy, S., Hogan, R. J., Hólm, E., Janisková, M., Keeley, S., Laloyaux, P., Lopez, P., Lupu, C., Radnoti, G., Rosnay, P., Rozum, I., Vamborg, F., Villaume, S., and Thépaut, J.: The ERA5 global reanalysis, Quarterly Journal of the Royal Meteorological Society, 146, 1999–2049, https://doi.org/10.1002/qj.3803, 2020.

Hooker, G., Mentch, L., and Zhou, S.: Unrestricted Permutation forces Extrapolation: Variable Importance Requires at least One More Model, or There Is No Free Variable Importance, Statistics and Computing, 31(6):82, http://arxiv.org/abs/1905.03151, 2021.

Hoyer, S. and Hamman, J. J.: xarray: N-D labeled Arrays and Datasets in Python, Journal of Open Research Software, 5, https://doi.org/10.5334/jors.148, 2017.

Huang, Y., Siems, S. T., Manton, M. J., Protat, A., and Delanoë, J.: A study on the low-altitude clouds over the Southern Ocean using the DARDAR-MASK, Journal of Geophysical Research Atmospheres, 117, https://doi.org/10.1029/2012JD017800, 2012.

Hubanks, P., Pincus, R., Platnick, S., and Meyer, K.: Level-3 COSP Cloud Properties (MCD06COSP_L3) Combined Terra & Aqua MODIS Global Gridded Product for Climate Studies User Guide, Tech. rep., NASA, https://atmosphere-imager.gsfc.nasa.gov/sites/default/files/ModAtmo/documents/L3_MCD06COSP_UserGuide_v13.pdf, 2020.

Humphries, R. S., Keywood, M. D., Gribben, S., McRobert, I. M., Ward, J. P., Selleck, P., Taylor, S., Harnwell, J., Flynn, C., Kulkarni, G. R., Mace, G. G., Protat, A., Alexander, S. P., and McFarquhar, G.: Southern Ocean latitudinal gradients of cloud condensation nuclei, Atmospheric Chemistry and Physics, 21, 12 757–12 782, https://doi.org/10.5194/acp-21-12757-2021, 2021.

Kay, J. E., Wall, C., Yettella, V., Medeiros, B., Hannay, C., Caldwell, P., and Bitz, C.: Global climate impacts of fixing the Southern Ocean shortwave radiation bias in the Community Earth System Model (CESM), Journal of Climate, 29, 4617–4636, https://doi.org/10.1175/JCLI-D-15-0358.1, 2016.

Kuma, P., McDonald, A. J., Morgenstern, O., Alexander, S. P., Cassano, J. J., Garrett, S., Halla, J., Hartery, S., Harvey, M. J., Parsons, S., Plank, G., Varma, V., and Williams, J.: Evaluation of Southern Ocean cloud in the HadGEM3 general circulation model and MERRA-2 reanalysis using ship-based observations, Atmospheric Chemistry and Physics, 20, 6607–6630, https://doi.org/10.5194/acp-20-6607-2020, 2020.

Lee, L. A., Pringle, K. J., Reddington, C. L., Mann, G. W., Stier, P., Spracklen, D. V., Pierce, J. R., and Carslaw, K. S.: The magnitude and causes of uncertainty in global model simulations of cloud condensation nuclei, Atmospheric Chemistry and Physics, 13, 8879–8914, https://doi.org/10.5194/acp-13-8879-2013, 2013.

Leinonen, J., Lebsock, M. D., Oreopoulos, L., and Cho, N.: Interregional differences in MODIS-derived cloud regimes, Journal of Geophysical Research, 121, 11 648–11 665, https://doi.org/10.1002/2016JD025193, 2016.

Lundberg, S. and Lee, S.-I.: A Unified Approach to Interpreting Model Predictions, in: 31st Conference on Neural Information Processing System, Long Beach, CA, USA, http://arxiv.org/abs/1705.07874, 2017.

Lundberg, S. M., Erion, G., Chen, H., DeGrave, A., Prutkin, J. M., Nair, B., Katz, R., Himmelfarb, J., Bansal, N., and Lee, S.-I.: From local explanations to global understanding with explainable AI for trees, Nature Machine Intelligence, 2, 56–67, https://doi.org/10.1038/s42256-019-0138-9, 2020.

Ma, J., Xie, S. P., and Kosaka, Y.: Mechanisms for tropical tropospheric circulation change in response to global warming, Journal of Climate, 25, 2979–2994, https://doi.org/10.1175/JCLI-D-11-00048.1, 2012.

Mace, G. G. and Protat, A.: Clouds over the Southern Ocean as observed from the R/V investigator during CAPRICORN. Part I: Cloud occurrence and phase partitioning, Journal of Applied Meteorology and Climatology, 57, 1783–1803, https://doi.org/10.1175/JAMC-D-17-0194.1, 2018.

Mace, G. G., Protat, A., and Benson, S.: Mixed-Phase Clouds Over the Southern Ocean as Observed From Satellite and Surface Based Lidar and Radar, Journal of Geophysical Research: Atmospheres, 126, https://doi.org/10.1029/2021JD034569, 2021a.

Mace, G. G., Protat, A., Humphries, R. S., Alexander, S. P., McRobert, I. M., Ward, J., Selleck, P., Keywood, M., and McFarquhar, G. M.: Southern Ocean Cloud Properties Derived From CAPRICORN and MARCUS Data, Journal of Geophysical Research: Atmospheres, 126, https://doi.org/10.1029/2020JD033368, 2021b.

Mallet, M. D., Alexander, S. P., Protat, A., and Fiddes, S. L.: Reducing Southern Ocean Shortwave Radiation Errors in the ERA5 Reanalysis with Machine Learning and 25 Years of Surface Observations, Artificial Intelligence for the Earth Systems, 2, 1–42, https://doi.org/10.1175/AIES-D-22-0044.1, 2023.

Mann, G. W., Carslaw, K. S., Spracklen, D. V., Ridley, D. A., Manktelow, P. T., Chipperfield, M. P., Pickering, S. J., and Johnson, C. E.: Description and evaluation of GLOMAP-mode: a modal global aerosol microphysics model for the UKCA composition-climate model, Geosci. Model Dev, 3, 519–551, https://doi.org/10.5194/gmd-3-519-2010, 2010.

Mason, S., Fletcher, J. K., Haynes, J. M., Franklin, C., Protat, A., and Jakob, C.: A hybrid cloud regime methodology used to evaluate Southern Ocean cloud and shortwave radiation errors in ACCESS, Journal of Climate, 28, 6001–6018, https://doi.org/10.1175/JCLI-D-14-00846.1, 2015.

McCluskey, C. S., Gettelman, A., Bardeen, C. G., DeMott, P. J., Moore, K. A., Kreidenweis, S. M., Hill, T. C., Barry, K. R., Twohy, C. H., Toohey, D. W., Rainwater, B., Jensen, J. B., Reeves, J. M., Alexander, S. P., and McFarquhar, G. M.: Simulating Southern Ocean Aerosol and Ice Nucleating Particles in the Community Earth System Model Version 2, Journal of Geophysical Research: Atmospheres, 128, https://doi.org/10.1029/2022JD036955, 2023.

McCoy, D. T., Burrows, S. M., Wood, R., Grosvenor, D. P., Elliott, S. M., Ma, P. L., Rasch, P. J., and Hartmann, D. L.: Natural aerosols explain seasonal and spatial patterns of Southern Ocean cloud albedo, Science Advances, 1, https://doi.org/10.1126/sciadv.1500157, 2015.

McDonald, A. J., Cassano, J. J., Jolly, B., Parsons, S., and Schuddeboom, A.: An automated satellite cloud classification scheme using self-organizing maps: Alternative ISCCP weather states, Journal of Geophysical Research, 121, 009–13, https://doi.org/10.1002/2016JD025199, 2016.

McFarquhar, G. M., Bretherton, C. S., Marchand, R., Protat, A., DeMott, P. J., Alexander, S. P., Roberts, G. C., Twohy, C. H., Toohey, D., Siems, S., Huang, Y., Wood, R., Rauber, R. M., Lasher-Trapp, S., Jensen, J., Stith, J. L., Mace, J., Um, J., Järvinen, E., Schnaiter, M., Gettelman, A., Sanchez, K. J., McCluskey, C. S., Russell, L. M., McCoy, I. L., Atlas, R. L., Bardeen, C. G., Moore, K. A., Hill, T. C. J., Humphries, R. S., Keywood, M. D., Ristovski, Z., Cravigan, L., Schofield, R., Fairall, C., Mallet, M. D., Kreidenweis, S. M., Rainwater, B., D'Alessandro, J., Wang, Y., Wu, W., Saliba, G., Levin, E. J. T., Ding, S., Lang, F., Truong, S. C. H., Wolff, C., Haggerty, J., Harvey, M. J., Klekociuk, A. R., and McDonald, A.: Observations of Clouds, Aerosols, Precipitation, and Surface Radiation over the Southern Ocean: An Overview of CAPRICORN, MARCUS, MICRE, and SOCRATES, Bulletin of the American Meteorological Society, 102, E894–E928, https://doi.org/10.1175/BAMS-D-20-0132.1, 2021.

Oreopoulos, L., Cho, N., Lee, D., Kato, S., and Huffman, G. J.: An examination of the nature of global MODIS cloud regimes, Journal of Geophysical Research: Atmospheres, 119, 8362–8383, https://doi.org/10.1002/2013JD021409, 2014.

Oreopoulos, L., Cho, N., Lee, D., and Kato, S.: Radiative effects of global MODIS cloud regimes, Journal of Geophysical Research: Atmospheres, 121, 2299–2317, https://doi.org/10.1002/2015JD024502, 2016.

Pedregosa, F., Varoquaux, G., Gramfort, A., Michel, V., Thirion, B., Grisel, O., Blondel, M., Prettenhofer, P., Weiss, R., Dubourg, V., Vanderplas, J., Passos, A., Cournapeau, D., Brucher, M., Perrot, M., and Duchesnay, E.: Scikit-learn: Machine Learning in Python, Journal of Machine Learning Research 12, 12, 2825–2830, 2011.

Pei, Z., Fiddes, S. L., French, W. J. R., Alexander, S. P., Mallet, M. D., Kuma, P., and McDonald, A.: Assessing the cloud radiative bias at Macquarie Island in the ACCESS-AM2 model, Atmospheric Chemistry and Physics, 23, 14 691–14 714, https://doi.org/10.5194/acp-23-14691-2023, 2023.

Pincus, R., Platnick, S., Ackerman, S. A., Hemler, R. S., and Patrick Hofmann, R. J.: Reconciling simulated and observed views of clouds: MODIS, ISCCP, and the limits of instrument simulators, Journal of Climate, 25, 4699–4720, https://doi.org/10.1175/JCLI-D-11-00267.1, 2012.

Platnick, S., Meyer, K. G., King, M. D., Wind, G., Amarasinghe, N., Marchant, B., Arnold, G. T., Zhang, Z., Hubanks, P. A., Holz, R. E., Yang, P., Ridgway, W. L., and Riedi, J.: The MODIS Cloud Optical and Microphysical Products: Collection 6 Updates and Examples From Terra and Aqua, IEEE Transactions on Geoscience and Remote Sensing, 55, 502–525, https://doi.org/10.1109/TGRS.2016.2610522, 2017.

Protat, A., Schulz, E., Rikus, L., Sun, Z., Xiao, Y., and Keywood, M. D.: Shipborne observations of the radiative effect of Southern Ocean clouds, Journal of Geophysical Research: Atmospheres, 122, 318–328, https://doi.org/10.1002/2016JD026061, 2017.

Rasp, S., Pritchard, M. S., and Gentine, P.: Deep learning to represent subgrid processes in climate models, Proceedings of the National Academy of Sciences, 115, 9684–9689, https://doi.org/10.1073/pnas.1810286115, 2018.

Regayre, L. A., Schmale, J., Johnson, J. S., Tatzelt, C., Baccarini, A., Henning, S., Yoshioka, M., Stratmann, F., Gysel-Beer, M., Grosvenor, D. P., and Carslaw, K. S.: The value of remote marine aerosol measurements for constraining radiative forcing uncertainty, Atmospheric Chemistry and Physics, 20, 10 063–10 072, https://doi.org/10.5194/acp-20-10063-2020, 2020.

Regayre, L. A., Deaconu, L., Grosvenor, D. P., Sexton, D. M., Symonds, C., Langton, T., Watson-Paris, D., Mulcahy, J. P., Pringle, K. J., Richardson, M., Johnson, J. S., Rostron, J. W., Gordon, H., Lister, G., Stier, P., and Carslaw, K. S.: Identifying climate model
structural inconsistencies allows for tight constraint of aerosol radiative forcing, Atmospheric Chemistry and Physics, 23, 8749–8768, https://doi.org/10.5194/acp-23-8749-2023, 2023.

Schuddeboom, A., McDonald, A. J., Morgenstern, O., Harvey, M., and Parsons, S.: Regional Regime-Based Evaluation of Present-Day General Circulation Model Cloud Simulations Using Self-Organizing Maps, Journal of Geophysical Research: Atmospheres, 123, 4259–4272, https://doi.org/10.1002/2017JD028196, 2018.

Schuddeboom, A. J. and McDonald, A. J.: The Southern Ocean Radiative Bias, Cloud Compensating Errors, and Equilibrium Climate Sensitivity in CMIP6 Models, Journal of Geophysical Research: Atmospheres, 126, https://doi.org/10.1029/2021JD035310, 2021.

Shapley, L. S.: 17. A Value for n-Person Games, in: Contributions to the Theory of Games (AM-28), Volume II, Princeton University Press, https://doi.org/10.1515/9781400881970-018, 1953.

Tselioudis, G., Rossow, W., Zhang, Y., and Konsta, D.: Global weather states and their properties from passive and active satellite cloud
retrievals, Journal of Climate, 26, 7734–7746, https://doi.org/10.1175/JCLI-D-13-00024.1, 2013.

Tselioudis, G., Rossow, W. B., Jakob, C., Remillard, J., Tropf, D., and Zhang, Y.: Evaluation of Clouds, Radiation, and Precipitation in CMIP6 Models Using Global WeatherStates Derived from ISCCP-H Cloud Property Data, Journal of Climate, pp. 1–42, https://doi.org/10.1175/JCLI-D-21-0076.1, 2021.

Vandal, T., Kodra, E., and Ganguly, A. R.: Intercomparison of machine learning methods for statistical downscaling: the case of daily and
extreme precipitation, Theoretical and Applied Climatology, 137, 557–570, https://doi.org/10.1007/s00704-018-2613-3, 2019.

Varma, V., Morgenstern, O., Field, P., Furtado, K., Williams, J., and Hyder, P.: Improving the Southern Ocean cloud albedo biases in a general circulation model, Atmospheric Chemistry and Physics, 20, 7741–7751, https://doi.org/10.5194/acp-20-7741-2020, 2020.

Varma, V., Morgenstern, O., Furtado, K., Field, P., and Williams, J.: Introducing Ice Nucleating Particles functionality into the Unified Model and its impact on the Southern Ocean short-wave radiation biases, Atmospheric Chemistry and Physics Discussions,
https://doi.org/10.5194/acp-2021-438, 2021.

Vergara-Temprado, J., Murray, B. J., Wilson, T. W., O'Sullivan, D., Browse, J., Pringle, K. J., Ardon-Dryer, K., Bertram, A. K., Burrows, S. M., Ceburnis, D., Demott, P. J., Mason, R. H., O'Dowd, C. D., Rinaldi, M., and Carslaw, K. S.: Contribution of feldspar and marine organic aerosols to global ice nucleating particle concentrations, Atmospheric Chemistry and Physics, 17, 3637–3658, https://doi.org/10.5194/acp-17-3637-2017, 2017.

Vergara-Temprado, J., Miltenberger, A. K., Furtado, K., Grosvenor, D. P., Shipway, B. J., Hill, A. A., Wilkinson, J. M., Field, P. R., Murray, B. J., and Carslaw, K. S.: Strong control of Southern Ocean cloud reflectivity by ice-nucleating particles, Proceedings of the National Academy of Sciences of the United States of America, 115, 2687–2692, https://doi.org/10.1073/pnas.1721627115, 2018.

Vignon, E., Alexander, S. P., DeMott, P. J., Sotiropoulou, G., Gerber, F., Hill, T. C. J., Marchand, R., Nenes, A., and Berne, A.: Challenging and Improving the Simulation of Mid-Level Mixed-Phase Clouds Over the High-Latitude Southern Ocean, Journal of Geophysical
Research: Atmospheres, 126, https://doi.org/10.1029/2020JD033490, 2021.

Walters, D., Baran, A. J., Boutle, I., Brooks, M., Earnshaw, P., Edwards, J., Furtado, K., Hill, P., Lock, A., Manners, J., Morcrette, C., Mulcahy, J., Sanchez, C., Smith, C., Stratton, R., Tennant, W., Tomassini, L., Van Weverberg, K., Vosper, S., Willett, M., Browse, J., Bushell, A., Carslaw, K., Dalvi, M., Essery, R., Gedney, N., Hardiman, S., Johnson, B., Johnson, C., Jones, A., Jones, C., Mann, G., Milton, S., Rumbold, H., Sellar, A., Ujiie, M., Whitall, M., Williams, K., and Zerroukat, M.: The Met Office Unified Model Global Atmosphere 7.0/7.1 and JULES Global Land 7.0 configurations, Geoscientific Model Development, 12, 1909–1963, https://doi.org/10.5194/gmd-12-1909-2019, 2019.

Watson-Parris, D., Rao, Y., Olivié, D., Seland, Nowack, P., Camps-Valls, G., Stier, P., Bouabid, S., Dewey, M., Fons, E., Gonzalez, J., Harder, P., Jeggle, K., Lenhardt, J., Manshausen, P., Novitasari, M., Ricard, L., and Roesch, C.: ClimateBench v1.0: A Benchmark for Data-Driven Climate Projections, Journal of Advances in Modeling Earth Systems, 14, https://doi.org/10.1029/2021MS002954, 2022.

Williams, K. D. and Webb, M. J.: A quantitative performance assessment of cloud regimes in climate models, Climate Dynamics, 33, 141–157, https://doi.org/10.1007/s00382-008-0443-1, 2009.

Yan, X., Liang, C., Jiang, Y., Luo, N., Zang, Z., and Li, Z.: A Deep Learning Approach to Improve the Retrieval of Temperature and Humidity Profiles from a Ground-Based Microwave Radiometer, IEEE Transactions on Geoscience and Remote Sensing, 58, 8427–8437, https://doi.org/10.1109/TGRS.2020.2987896, 2020.

Zhang, C., Zhuge, X., and Yu, F.: Development of a high spatiotemporal resolution cloud-type classification approach using Himawari-8 and CloudSat, International Journal of Remote Sensing, 40, 6464–6481, https://doi.org/10.1080/01431161.2019.1594438, 2019.

Zipfel, L., Andersen, H., and Cermak, J.: Machine-Learning Based Analysis of Liquid Water Path Adjustments to Aerosol Perturbations in Marine Boundary Layer Clouds Using Satellite Observations, Atmosphere, 13, 586, https://doi.org/10.3390/atmos13040586, 2022.