# Peer review of "A machine learning approach for evaluating Southern Ocean cloud-radiative biases in a global atmosphere model"

_EGUsphere, 2023_

## Author Comment (AC1)

**Response to Reviewer 1**

We wish to thank this Reviewer for their positive comments and their suggestions to expand our discussion. We have made more detailed comments below in this respect.

**Comment**

*I'm not sure if it belongs in the introduction, or discussion, but I am struck by the relationship between this approach and calibration approaches which use an emulator to relate model parameters to (potentially multiple) model outputs (c.f. Watson-Parris et al. 2022) in order to reduce model biases. Such emulators allow estimation of the sensitivity of particular outputs to given inputs (essentially feature importance, see e.g. Lee et al. 2013), and have recently been used with multiple observables (LWP, Nd, etc) to constrain ERF_aci (Regayre et al. 2023). I feel the authors' approach adds a useful step in relating the biases to observables, and this might help alleviate the difficulties in that work in choosing the 'best' observations to use for bias (or uncertainty) reduction. Generally, since the authors perform a small perturbation study I think it would be useful to more explicitly link the approach to model tuning and discuss how it can help improve the process.*

Thank-you for highlighting these studies. We have now included some comparison to the study presented in Watson-Parris et al. (2022) in our introduction. We have also been following the work of the perturbed parameter ensembles such as that of Regayre et al. (2021 and 2023) with interest, and as a group have discussed potential ways to extrapolate our methods into this space. We have now included some discussion on this topic in the introduction and conclusion.

Line 54: 'Another way to evaluate and in some instances tune models is to explore parameter uncertainty (Lee et al. 2013, Regayre et al. 2021, 2023). In these cases the parameter space (the range of plausible values) and their impacts in global climate models are emulated with more simplified statistical models. This allows re-sampling over a range of multi-parameter values many times over what is possible with physically driven models. From these large samples, the uncertainty attributed to particular parameters can be identified and the best combination of parameter values can be constrained based on comparisons with observations. These methods present a powerful way of reducing uncertainty of climate models within known and quantified parameters and physical mechanisms.'

Line 77:'Of particular note is the application of machine learning to climate emulation, i.e. emulating the global response of complex climate models, as outlined in [?]. Climate emulation has typically used simple models to estimate what the response of the climate (usually temperature), may be to changes in forcings. These models tend to not capture spatially varying and non-linear processes well, whereas machine learning has been shown to do well in this space, but has been challenged by a lack of data for training purposes. [?] have now provided a dataset and some initial machine learning frameworks designed specifically for training models for this application, which may provide a new way to determine possible climate responses to changes in forcings, beyond that of the temperature.'

Line 603: 'This finding is particularly relevant to methods used to constrain models to observationally plausible values, such as that done in Regayre et al. (2021, 2023). In these studies, perturbed parameter ensembles (PPEs) have been used to sample distributions of many parameters, after which, observations are used constrain the model to internally consistent and plausible values. These studies, which provide an efficient and comprehensive way to both evaluate and tune model parameters, use huge arrays of data representing complex changes in the model. While alternative methods to determine feature importance have been implemented in these studies, an approach such as the one presented in this work would provide an efficient way to interpret the effects of the parameter tuning. We further note recent developments in SHAP in which multiple targets can be predicted and evaluated, potentially providing a significant advantage for studies using PPEs.'

**Minor typos**

*L257: 'but' - 'by'*
We have fixed the above as suggested.

*L404: understating - understanding*
We have fixed the above as suggested.

---

## Author Comment (AC2)

**Response to Reviewer 2**

Many thanks to this Reviewer for their careful and detailed comments. We appreciate the time they have spent highlighting where our methods and logic were not clearly communicated or not enough information was provided. We believe we have now resolved the questions this Reviewer had around our methods, including that of the satellite products and COSP, as well as our choice of algorithm. We have also performed more analysis to demonstrate the advantages of the XGBoost/SHAP method over other methods. Our detailed responses can be found below.

**Overview:**

*I am entirely in favor of developing new approaches to evaluate climate models, and in particular, I find the additive nature of the SHAP values interesting. That said, I seem to be missing (or not correctly understanding) some key aspects, such as how the dependence between the "cloud features" (what one might call collinearity in a multiple linear regression) means for interpreting the results. As is, I don't perceive how this new approach is providing new insights beyond what existing approaches could provide. I explain in more detail in my comments that follow below. This doesn't make the results incorrect, and in fact, clearly the primary scientific result called out in the abstract – that biases in liquid water path are the largest contributor to the cloud radiative biases – is correct. In my view this article would be much stronger if it directly compared results based on cloud regimes and multiple linear regression (both of which seem to have already done at some level) and demonstrated in a concrete way the additional value of the XGBoost approach.*

We have performed some additional analysis in this respect, which is detailed below.

**General Comments:**

**1) On the relative merits of the XGBoost/SHAP technique presented here.**

*At several points in the manuscript, you assert that the XGBoost technique presented here achieves results that current techniques do not. For example, you write in the abstract that "most evaluation methods focus on specific synoptic or cloud type conditions and are unable to quantitatively define the impact of cloud properties on the radiative bias whilst considering the system as a whole." Or later on line 467, you write "No other analysis method is able to consider a system as a 'whole' (within the confines of the data provided that is), and subsequently isolate the effect of individual components."*

*Frankly, I do not see how these statements are correct. Nothing, for example, stops one from taking a "cloud regime" analysis, aggregating data by regimes, and calculating the mean SWCRE and LWP bias for each regime – with the result that one ends up know the contribution of each regime to the overall biases in SWCRE and LWP. Or indeed running a multiple-linear regression to map the SW bias (with or without cloud regimes) to some set cloud properties. In what way would such analyses be "not quantitative" and/or "not consider the system as a whole"?*

We take the reviewer's point here that there are methods that can achieve a similar aim as what is presented in this work, and in fact, our previous work has already addressed this to a degree (Fiddes et al. 2022, see Figures 8-10). The point that we make in this study is that it can be done in one step with our study's method, i.e. there is no need to separate data into cloud regimes to then later aggregate them. The method we presented can capture the known cloud specific relationships, without any knowledge of the cloud types themselves (granted - the definitions of the cloud types - eg optical depth and cloud top pressure are predictors, but weaker ones at that). Furthermore, and in response to other comments from the Reviewer, we now highlight the ability of this analysis to understand interactions, which few other methods can achieve. We have revised the statements identified to make these points more clear. We further make the point that we expect that this type of analysis may be of use outside of the Southern Ocean cloud-radiative bias problem (hence our submission to GMD), where datasets may not have well known or defined groupings.

Line 2: 'To date, most evaluation methods focus on specific synoptic or cloud type conditions that do not consider the entirety of the Southern Oceans cloud regimes at once. Furthermore, it is difficult

to directly quantify the complex and non-linear role that different cloud properties have on modulating cloud radiative effect. '

Line 534: 'Our method is able to easily account for individual and interaction, sometimes non-linear, influences from each predictor, while still allowing us to interpret our predictions are made.'

*Indeed, you appear to have already applied a multiple-linear regression (I think perhaps to the region as a whole rather than cloud regimes); where on line 208/218 you indicate that a multiple linear regression was "only" able to predict between 42-43% of the variance in SWCRE as compared to 58% with XGBoost. Frankly, I am not sure that I see going from 42 to 58% of explained variance as a major improvement. And (if I understand) the multiple-linear regression also identifies bias in LWP as the principal predictor for bias in SWCRE (line 200-203).*

*As a paper where one of the principal goals is to argue that the XGBoost/SHAP analysis is superior (in some aspects), I think it would be better to compare directly the XGBoost/SHAP analysis to other approaches and demonstrate in a concrete way (that goes beyond commenting on the explained variance) what is gained.*

We would argue that the improvement by XGBoost is meaningful, in particular because of the non-linearity that exist between the cloud predictors and the SWCRE, and the fact our analysis has shown that this method supports the body of knowledge in the literature. We would then ask, why would we continue to use an inferior method? To address your concerns, we have now included in the Appendix a heat map of the correlation values of each cloud predictor with the SWCRE, where the remaining predictors are shown to have very weak linear relationships with SWCRE (below 15%). Examination of these relationships visually indicates that non-linear processes dominate (we have included a plot of the mean values for each cloud type/location in the Appendix), contributing to our decision to not proceed with a multiple linear regression.

Line 245: 'Analysis of the mean $SWCRE_{TOA}$ bias versus the cloud feature biases for each of the cloud types developed by F22 over three latitudinal areas is shown in Figure 2. This figure confirms, even just considering the means across cloud types/locations, that these relationships are in some cases highly non-linear, and in other cases, very weak. '

We have explored more deeply the results from the multiple linear regression. In the first instance, the MLR performs worse across all fronts, with its median, mean and standard deviation all further away from the true values. We have also repeated Figure 1 for the MLR, now found in the Supplementary Material, which shows that the MLR tends have more overestimated values in regions of the strongest SWCRE bias, and the more underestimated values in the regions where the bias is less strong. The histogram shown in the Supplementary Figure 1d also indicates a less symmetrical relationship of the predicted bias with the residual, another indicator of poorer performance. Finally, we have performed a normalised MLR to evaluate the contribution of each predictor to the MLR results.

Supplementary Material :'The MLR was able to predict between 42-43% of the variance (when tested on different summers, in the same way as described for the XGBoost training and testing data sets), with a predicted mean of $12.4\,\mathrm{W\,m^{-2}}$, median of $13.0\,\mathrm{W\,m^{-2}}$ and a standard deviation of $29.2\,\mathrm{W\,m^{-2}}$, compared to mean of $12.4\,\mathrm{W\,m^{-2}}$, median of $11.7\,\mathrm{W\,m^{-2}}$ and standard deviation of $44.4\,\mathrm{W\,m^{-2}}$ for the true values. Figure 1 shows the true spatial $SWCRE_{TOA}$ bias (a), the MLR predicted bias (b), residual (c) and a histogram of the residual against the prediction (d). In this final subplot, a more symmetrical concentration of residuals (y-axis), centred around zero, and a narrow range of predictions is an indicator of a well performing model (x-axis). Here we can see that the values are skewed towards more negative negative values for the residual, while the prediction is more evenly distributed. This indicates that the model is tending to under predict the $SWCRE_{TOA}$ bias more frequently than it over predicts it, regardless of what the prediction value is.

While LWP has the largest linear relationship with the SWCRE (as shown in Appendix Figure A3), in a normalised multiple linear regression, the liquid water cloud optical depth (TauL) has the largest co-efficient, then LWP, IWP, CFI, CFL, CTP and TauI. This result is somewhat different to what the SHAP features indicate (as discussed in Section 4.1). Unlike with the SHAP analysis, we cannot then look at

how the individual contributions of each predictor contribute to the final result of individual or averaged predictions, limiting our ability to analyse this further without repeating this analysis across individual components. This is another reason why we think that the XGBoost/SHAP method presented is superior, given it can be modelled using the entire dataset, and then analysed for its individual components.

These results suggests that linear assumptions are less suitable for a problem such as this, and that weak-moderate multi-collinearity must be taken into account. Furthermore, beyond evaluation of the correlation values, mean statistics, and co-efficients, the MLR cannot then be further split into differing spatial, temporal (or both - e.g. cloud types) to further understand its predictions. Performing individual MLRs on each cloud regime (or even individual lat/lon cells even) can over come this to a degree, but this results in the compartmentalisation of a system, where we are looking for a method that can consider the system and its complexities as a whole.'

**2) Dependence between cloud "features"**

*I don't understand how the dependence between cloud variables / features is being taken into account in the XGBoost/SHAP analysis. Yes, I see you have a dendrogram and clustering index (Figure 2c), but these metrics don't seem to be used in interpreting later results.*

We have made sure to include the results of this analysis and the additional interaction analysis on the later results:

Line 336: 'We can further investigate the nature of feature interaction by using the SHAP interaction values. SHAP interaction values are similar to SHAP values, but provide the 'main' contribute from each feature (along the diagonal) plus the interaction values for each combination of features, for every point in the data set. How to interpret these values is explained in more detail in Section 2.3. In Figure 3 we show the absolute mean values across the whole data set. We note that the off-diagonal values show the total interaction between the two features (see Section 2.3 for more details on how this is calculated).

Figure 3 shows us that the LWP has the highest 'main' interaction, indicating that it provides the most important, individual contribution overall. This is then followed by the CFI, CFL, IWP, TauI, CTP and TauL, which is a slightly different order to that provided by the overall SHAP values in Figure 2c, though this is only occurring when SHAP values are very close in magnitude. Importantly, we can see that the absolute mean of the interaction values are overall, smaller than the main values, indicating that feature interaction is less important to the overall result than the main contributions. If we consider the off-diagonal values in Figure 3, we can see that the largest value is between LWP and IWP, though this value is still smaller than even the lowest 'main' SHAP value. We believe this analysis supports our earlier finding of only weak-moderate collinearity between cloud features and means that we are able to make inferences about how each cloud feature is impacting the radiative bias physically.'

Line 414: 'While our conclusions above make physical sense, the SHAP values presented do not tell us about how individual pairs of features may interact with each other to provide these results. Our linear regressions and clustering analysis has suggested that each of these features have little dependence on one another, however, our knowledge of the physical world would suggest that some interaction exists. To explore this further we use the SHAP interaction values, which provide a quantitative value of the interaction, as well as the 'main' component from an individual feature alone. As Figure 3 indicated, the main SHAP component for each feature is the dominant driver of the overall SHAP values. We have explored this further by reproducing Figure 4, but for the main component alone. This plot is shown in the Appendix (Figure A4) and shows very little difference to that of Figure 4 in terms of pattern, but does show slightly increased SHAP values. With this result we can be confident that the relationships we are deriving are attributable to the individual feature, and that the interactions between features do not make up the majority of the 'information' provided. . '

*For example, TauL is obviously related to LWP and CFL. What are the panels f, m, and t in Figure 3 showing me? If I understand, the SHAP values in Figure 3f represent only the "additional" information in the tau-bias that is not covered LWP and CFL? (In short, it is not a measure of how much tau errors contribute to the biases, but rather the contribution of using TauL being used as an additional correction factor on top of LWP and CFI). If yes, does this mean factors such as effective radius OR the inherent*

*non-linearity between LWP and albedo/TOA SW flux is being "modeled" by this term?*

Your interpretation is correct in that the SHAP values represent the additional change in prediction from the XGBoost model by including that particular feature. The SHAP values attributed to the any feature is information that cannot be provided by the other features, but it may have some inherent dependence on them. With this in mind, and in relation to the comment above, we have used the SHAP interaction feature analysis to ensure that our conclusions are robust: i.e the relationships we are deriving from the SHAP values are attributed primarily to the feature of interest. We can also explore in more detail particular relationships, such as the TauL-LWP or TauL-CFL relationships.

The TauL bias has a very weak linear correlation to both LWP and CFL ($R^2$ values of 0.14 and 0.02 - see Figure A3). While our correlations and clustering analysis suggest that our features are relatively independent, we can explore this further to try to make physical sense of the results as you are suggesting. We have produced interaction values for each pair of predictors, in addition to a 'main' component for each predictor, which, when summed together for any individual point, will equal the original SHAP value for that point. We can use this information to see that, in the case of TauL and LWP or TauL and CFL, the absolute mean interaction values are indeed important relative to the 'main' values (See Figure 3) and we would suggest that this is capturing some of the non-linearities for TauL, LWP and effective radius for example.

*Somewhat similarly, I am not clear on what the "offset" in the CFL (Figure 3p) - or the offset in other variables means. As-is, the situation appears no different to me than the interpretation of variables in a multiple-linear regression, and is superior in the sense that linearity is not assumed. (Indeed the bottom row suggests that the relationships are not linear, at least for larger biases in the cloud features, which is no doubt how the explain variance can go up, but this would – if anything- make the interpretation more difficult).*

The reviewer is correct that the relationships are non-linear. We have edited the text to make this clearer, as well as clarified the role of the base value prediction.

Line 374: 'While this makes sense when considered together with the base value, this result still demonstrates that the radiative influence of clouds is not as simple as "less cloud (even marginally) equals more sunlight passing through", but highlights the ability of the SHAP analysis to capture non-linear processes. '

*I must admit that I found myself wonder what would happen if you put SW cloud albedo into you set of cloud features?*

We had also considered something similar to this, and tested it with all-sky SW radiation. When we did this, the skill of the model increases dramatically. However, we also felt that it took the focus away from trying to understand the role of cloud biases in driving the radiation bias. In effect, if we wanted to build a model with the solely the best predictive skills, using what ever fields we could, then we could have definitely have built one with greater skill than what we have presented. But that was not the aim of this work - the aim was to understand the contribution of cloud fields to the radiative bias, and hence we have 'settled' on a less skilful model, but one that can provide us with the information we are after.

**3) Early description, cloud "features" & data**

*I initially found much of the early text difficult to follow, in part because of the abstract language and use of the broad term "cloud features". It is not until Figure 2 (on about page 9, well into the results section) that I understood that this meant LWP, CFL .... I think it would be helpful to introduce the "features" you are going to use in section 2.1.*

We have now been more explicit in our language at the beginning of the methods section.

Line 156: 'In this work, we use XGBoost to predict the $\text{SWCRE}_{TOA}$ biases using the biases in the cloud properties described in the previous section. We refer to these cloud properties as 'features', in line with the language used in machine learning.'

*Please also see a variety of specific comments as regards these data.*

Thank-you for this - we have addressed the specific comments below.

*As is, I am still not clear as to whether LWP is referring to "in cloud" LWP or "domain mean" LWP (meaning a spatial average which includes clear sky / non cloudy "pixels" in a given spatial domain). And the same is true of IWP, TauL and TauI.*

We have now clarified that the these cloud properties refer to the domain mean, or what we refer to as the grid box mean. In the paper that preceded this, Fiddes et al. (2022), we outlined the reasoning for this choice and the methods used to ensure we are comparing like for like. We refer the Review and our readers to this previous paper for more detail.

Line 133: 'The cloud fields of interest include the grid box mean liquid and ice cloud fractions (CFL and CFI), liquid and ice cloud optical depths (TauL and TauI), and cloud top pressure (CTP). These are described in detail in F22, including the pre-processing performed and the decision making around what specific data set to use.'

**Specific Comments:**

*Line 27. You write "Currently, our climate and weather models do not take into account the pristine composition of the SO atmosphere . . . ". What constitutes "our" models? Perhaps "some" or "many" and provide references?*

We have made the change to 'many' as suggested and provided references.

Line 27: 'Currently, many climate and weather models do not take into account the pristine composition of the SO atmosphere, assuming, like over much of the world, that INP are available to help freeze cloud droplets, resulting in too many ice-phase clouds, which allow too much shortwave radiation to reach the surface of the ocean (Vergara-Temprado et al. 2018; McCluskey et al. 2023).'

*Line 97. COSP contains a variety of simulators. Do you mean just the ISCCP or MODIS simulator? Please clarify.*

We confirm that we are using the MODIS simulator.

Line 112: 'Importantly, for this work we have the COSP simulator switched on (Bodas-Salcedo et al. 2011), in this case for the the Moderate Resolution Imaging Spectroradiometer (MODIS) satellite, to allow for sensible comparison between satellite fields and the model.'

*\*Line 103. It is good that you are identifying the satellite products, but I think you should go a step further and identify which specific "variables" or "fields" are being used, and discuss implication / limitation (see also comment line 114). With MODIS, for example, not every pixel has a successful retrieval. (I know you are using the L3 summary product, but this product still represents a subset of observed pixels). In particular, you can find LWP is both the MODIS and CERES products you list. \*What LWP variable did you use? \*In principle, you should include some discussion on the quality of the observational data, and discuss any limitation that might affect your analysis.*

We thank the reviewer for their concern here. To clarify, we are using only MODIS cloud properties, and only CERES radiative fields. We have more explicitly stated this now:

Line 119: 'We use two satellite products in this work: cloud properties from MODIS Combined Aqua/Terra, Level 3 daily, 1x1 degree grid, Collection 6.1, COSP product (MCD06COSP_D3_MODIS) derived specifically for CMIP6 (Pincus et al. 2012; Platnick et al. 2017 and Hubank et al. 2020) and radiation fields from the Clouds and the Earth's Radiant Energy System (CERES) Syn1Deg product (Doelling et al. 2013 and 2016).'

With respect to a discussion about the quality and limitations of these retrievals, we direct the Reviewer to our preceding paper, Fiddes et al. (2022), where we have spent some time on this in Section 2.2. Given this current paper uses the exact same variables and processing, we have not repeated this discussion. We have now made clear that we have considered these points previously and refer readers to our earlier work for more information.

Line 123: 'How these products have been prepared is fully described in F22, which has used the exact data set as this current work. F22 includes discussion about the satellite products strengths and limitations, quality (including successful pixel retrieval), past evaluation and processing. '

*Line 107. Please give the ACCESS-AM2 horizontal grid resolution?*

We have provided this in the previous paragraph on Line 117

*\*Line 114. You write "... the model's COSP liquid water path (LWP) and ice water path (IWP) showed considerable biases when compared to the observed COSP products, and hence the raw model output was used for these fields instead." I am not clear why this justifies using the raw model output! Is not the point of the simulator to make satellite datasets comparable to model? For example, you don't know from MODIS observations the cloud LWP when deep cloud systems are present with lots of ice condensate at the top. You only get an IWP retrieval for MODIS. But your model will have LWP present in the deep systems.*

We understand the reviewer's concern in this respect and emphasis that we did not make this decision lightly. We refer to a section in our previous work (Fiddes et al. 2022):

"Very large biases that we considered unrealistic were found for the modelled Reff, and hence these fields have not been used for this work. Similar biases were found for the LWP and IWP, and hence, the COSP-derived products (described in the next section) for these fields were not used but were replaced with the raw model output. While this adds a degree of uncertainty to this work, we believe such an analysis with the derived COSP fields would not have been useful."

"As discussed in Sect. 2.2, significant issues remain with the retrievals of Reff and subsequently the water paths. The propagation of these errors through the COSP framework meant that comparisons between the COSP product and MODIS product were found to be very unrealistic, and hence these fields were not used and the raw model field LWP and IWP were used instead, adding uncertainty to these results."

You may also read our response to Reviewers on this topic for F22, (https://acp.copernicus.org/articles/22/14603/2022 22-14603-2022-discussion.html) which includes a link to analysis showing the quality of the COSP retrievals for these fields.

Ultimately, we acknowledge that this decision brings with it considerable uncertainty, but we had much less confidence in the COSP derived products in this instance than the raw modelled fields. We have made further note of this now in the methods as well as in our discussion section.

Line 137: 'This is thought to be a continuation of poor retrievals of the cloud effective radius. While we acknowledge that this bring uncertainty into our results, we have greater confidence in the raw model fields in this instance. For this reason the raw model output was used for these fields (LWP and IWP) instead.'

Line 554: 'We note that the cloud fields used in this work, including the satellite products, and the modelled products each contain inherent uncertainties. While the MODIS L3 product has specifically been produced for model evaluation, we must acknowledge that these products may not represent the 'truth'. Greater discussion on this can be found in F22. Similarly, Pei et al. (2023) find an underestimation of short wave cloud radiative effect at the surface of $7.9\,W\,m^2$ in CERES compared to ground observations at Macquarie Island, indicating that similar issues exist in the satellite radiative fields. To add to this, despite satellite simulators such as COSP being designed to reduce the uncertainties between modelled and satellite retrieved products, we have found that some of these simulated fields were of too

poor a quality to be used with confidence. This was the case for the LWP and IWP fields, for which we used the raw modelled products instead. While this decision adds to the uncertainty of our analysis, we are none-the-less confident in our overall results (eg. LWP being a dominant driver of biases radiative biases).'

*Also, as far as I know there is no simulator built specifically for the CERES-SYN product. I am guessing here you are using MODIS data for cloud "features" and CERES SYN only for radiation/SWCRE, Yes?*

Yes, we have clarified that we are only using COSP for the cloud properties (aka features), which are coming from MODIS. See our previous response on this point.

*Line 119. What does "fitted" mean here? Please expand this description.*

In this instance, we used the MODIS cloud fields to develop cloud types using k-means clustering. We then took the resultant cluster centers, and 'fit' the ACCESS data to these, i.e. find which cluster centre was the closes to each data point in the respective ACCESS data set. We have expanded on this in the text as suggested:

Line 141: 'These cloud types were developed using $k$-means clustering, where 12 cloud types were found using the MODIS data set. The 12 cluster centers defined by k-means were then applied to the respective ACCESS-AM2 product, so that each data point was assigned the cluster (aka cloud type) that most closely fit.'

*Line 136. Using the word 'folds' to describe "4-Fold cross validation" is not very helpful. What does a "fold" entail? Please expand/rephrase this description.*

k-fold (in our case 4-fold) cross validation is a commonly used machine learning term which we have now explained more explicitly in the text.

Line 161: 'We have used 4-fold cross validation which splits the training data into four individual data sets, in effect generating an ensemble. We note that the 'folds' did not split the data at random, but rather into continuous sections in time, so to avoid the risk of over fitting due to auto correlation.'

*Figure 1. Please provide area-weighted mean values for the thee maps on the left. What is the spatial scale of the data (is this a fixed lat/lon grid)? What year (or years) of data is this, all 5 years?*

We have now provided the area weighted mean values as suggested.

Line 268: 'The area weighted statistics for the entire region for the true and XGBoost predicted values respectively are: means of $12.0 \, \mathrm{W\,m^{-2}}$ and $11.5 \, \mathrm{W\,m^{-2}}$; medians of $11.1 \, \mathrm{W\,m^{-2}}$ and $10.6 \, \mathrm{W\,m^{-2}}$ and standard deviations of $45.2 \, \mathrm{W\,m^{-2}}$ and $33.5 \, \mathrm{W\,m^{-2}}$'

The spatial scale of the data is from 30-69S on a regular lat/lon grid, which we have also now included in the methods section. The data shown in this plot, and for the remainder of the analysis is for the full 5 years. We have also made this more clear.

Line 132: 'Our analysis has been limited to the region of 30-69°S.'

Line 174: 'For the following methods and analysis, we have run the tuned XGBoost model over the entire data set. While this may lead to some over fitting (up to 3% of explained variance), we felt it was important to capture some year-to-year variability, as opposed to just one summers worth of data. '

*Figure 1. The scatter plot does not seem very useful. What are the units on the colorbar? Perhaps replace with plot showing absolute bias vs percentage of data. That is, if you take the X% of points with the smallest (absolute value of) of bias, what is the maximum value of the (absolute) bias of this subset. Plot maximum value of the bias vs. X %. A non-linear scale might help here (for example, X = 100, 99, 95, 85, 75, 50, 25, 5, 1).*

We thank the reviewer for this suggestion. We have now replaced the plot with a plot showing the density of the residual verses the predicted bias, which we believe highlights the performance of the XGBoost models. We have now included the following text:

Line 270: 'In Figure 1d, a more symmetrical concentration of residuals (y-axis), centred around zero, and a narrow range of predictions is an indicator of a well performing model (x-axis). We can see that the XGBoost model does provide a relatively symmetric pattern, with little skew in any direction. This is especially the case when compared to the MLR, shown in the supplementary material. '

*Figure1 Caption. (a) Are all biases given in this article (obs - model)? Please define and use consistently. I think perhaps you have this backwards in the caption. It is typical to use "model – obs" such that a negative bias means the model value is too small (as compared to the observations) – And I think later in the article negative LWP bias does mean model LWP is too small (as compared to observations). (b) You should also define how you are calculating SWCREtoa in the text (I suspect this might explain the apparent sign reversal here).*

Yes, you are correct here - we had the caption backwards. All biases have been calculated as model - obs, in line with convention. We have corrected this in the caption as suggested.

We have also now included in the results section a reminder (it was defined in the methods section) of how the CRE is calculated for clarity.

Line 223: 'As a reminder, the $SWCRE_{TOA}$ is calculated as the clear-sky radiation minus the all-sky radiation fields, which results in negative $SWCRE_{TOA}$ values (see Figure A1). The biases are calculated as model minus observations, with positive $SWCRE_{TOA}$ biases indicating that the model has less negative values than the observations, resulting in a positive $SWCRE_{TOA}$ bias, indicating that less sunlight is being refelted out to space.'

*Line 187. You write "Examination of the model and satellite fields separately (not shown) shows that that the asymmetrical bias appears to be due to ACCESS-AM2 failing to capture the observed spatial variability." This seems like an interesting (and to me surprising) point. Perhaps show the individual fields in figure 1, not just the differences?*

We have now included these plots in the Appendix of this paper, though they do not show this asymmetry as clearly. We also found this quite interesting, and noted that the only cloud feature the captured any similar asymmetry with respect to the SHAP values was the cloud top pressure. However, the residual shown in Figure 1 indicates that even our XGBoost model does not really capture this aspect of the bias, indicating to us that the biases in cloud features may not be the leading cause. We have added a note in this section, that this is discussed in more detail at a later stage in the article.

Line 232: 'We will consider causes of this asymmetry again in Section 4.2.'

*Line 184. "... reaching the surface.."? Here and at several other points in the text you refer to "too much reaching the surface". While I have no doubt this is true, you are using the CRE defined at TOA which means "more SW absorbed in the system" not just more SW reaching the surface. I know this is nitpicking, but I think it would be more technically correct to stick with "too little SW being reflected back to space" or "too much absorbed SW."*

Thank-you for raising this subtlety - we have revised the manuscript to ensure we use the more correct 'too little SW being reflected back to space'. For this particular sentence, we have changed to:

Line 224: '... corresponding to too little shortwave solar radiation being reflected back out to space by clouds, and too much being absorbed into the Earth system, including reaching the surface of the ocean.'

*Line 209. How does the bias and standard deviation in the prediction for bias from the multiple-linear regression compare with those in the next paragraph (lines 221-223) given for XGBoost?*

The MLR generally under predicts the standard deviation to a greater degree. We have added this to the text of the supplementary material:

Line 2 (supplementary material): 'The MLR was able to predict between 42-43% of the variance (when tested on different summers, in the same way as described for the XGBoost training and testing data sets), with a predicted mean of 12.4 W m2 , median of 13.0 W m2 and a standard deviation of 29.2 W m2 , compared to 12.4 W m2 , 11.7 W m2 and 44.4 W m2 for the true values. '

*Figure 2c. The dendrogram is very hard to read. The vertical grey lines are too close together.*

We acknowledge that the dendrograms lines are close together but we suggest that this is the key point of this figure, that the lines are close together. We have revised the figure so that the clustering cut-off is now at 0.8, which better indicates the height at which the clusters are merged.

*Line 280. You write, "A distance of one would imply feature independence, while zero would imply complete redundancy." So if I understand, the lines have very similar clustering distances and are relatively close to one. Does this mean each variables contain largely equal amounts of information? Please discuss what this means as regards the relationship between the specific "features" you have choosen. For example, what does this mean as regards understanding what TauL, LWP, and CFL mean? Please see general comments.*

The SHAP clustering analysis measures feature redundancy by performing model loss comparisons and then uses hierarchical clustering to measure how similar their impact is on the end result. Having similar clustering distances does not imply that each variable provides equal amounts of information to the SHAP analysis, this is demonstrated by the bar plot, rather, it is a measure of independence of information. Our clustering analyses indicates that our features are providing largely independent information, which is supported by the lack of strong correlations between the features. We have altered the text as follows:

Line 332: 'Here we see that the least important cloud feature (CTP) is the first to be merged into clusters with the most important cloud features (eg. CFI). However, the merge is occurring only slightly before other features are merged into the other clusters, indicating that even the weakest cloud features are providing independent, if not as important, information to the XGBoost model.'

*Line 309. I think "CFI" on this line should be "CFL".*

We have corrected as suggested

*Line 314. I am struggling to understand what this "offset" DOES mean. All other cloud properties being fixed, less cloud DOES mean more sunlight would reach the surface. So is this telling me there are compensating errors (for example having too much LWP is being compensated for by too little CFL) OR is this a consequence of non-linearity in the system (for example, on average LWP is larger on average when CFL is larger) OR simply radiation is not a linear function of LWP)? As written, you seem to be telling me what it does NOT mean, and I would rather understand what is DOES mean.*

We suggest, as you do, that the offset is a representation of the SHAP-analysis highlighting non-linear processes. For example, when the CFL bias is weakly negative, it is associated with weakly negative SHAP values. These weakly negative CFL biases occur predominantly in the sub-polar region, where the LWP bias is still positive, potentially leading to a case where clouds are more optically thick and hence more reflective, resulting in the negative SHAP value. We have altered this last sentence to reflect this:

Line 374: 'While this makes sense when considered together with the base value, this result still demonstrates that the radiative influence of clouds is not as simple as "less cloud (even marginally) equals more sunlight passing through", but highlights the ability of the SHAP analysis to capture non-linear processes. '

*Line 317. I think the phrase at the end of the sentence which includes the word "compensating" is potentially confusing. I think you mean mid-latitudes and high-latitudes bias are compensating in the hemispheric mean, but maybe you mean CFI biases are compensating something else (CFL biases)? Perhaps rephrase for clarity.*

We have removed the second part of this sentence for clarity.

Line 378: 'Weakly negative biases in CFI correspond to moderately strong positive SHAP values in the mid-latitudes, while positive CFI biases contribute to moderately negative SHAP values.'

*Line 322. "not intuitive"? The CFI and IWP features aren't "ice in place of liquid"; It is simply too much or too little ice (regardless of what liquid is doing). The features make intuitive sense to me as positive bias in CFI and IWP (inner two rings) gives a negative SHAP factor just as too much LWP gives a negative shape factor – and vice versa the negative bias in CFI and IWP in the (outer ring) gives a positive shape factor. If there is ice in place of liquid than liquid bias will necessarily be negative when ice bias is positive – that is you can't look at these terms in isolation to see the phase change / compensating errors – and indeed this is occurring and you do comment on this in the sentences that follow.*

Thank you for raising this concern - we have revised the sentence as follows:

Line 383:'Negative SHAP values are associated with positive IWP and CFI values where too much ice is resulting in too much $SWCRE_{TOA}$ being reflected out to space.'

*(The only thing that is not intuitive to me is that a negative SHAP factor – due to increase in LWP or IWP – means a "negative radiation bias" (line 300). But if I understand this is due to the way SWCRE_TOA is being defined).*

We have included a sentence on how the CRE should be interpreted.

Line 127: 'We have defined the $SWCRE_{TOA}$ as the difference between the clear-sky radiation and the all-sky radiation fields (for both the model and the satellite products). A positive $SWCRE_{TOA}$ bias indicates that the ACCESS-AM2 model is allowing too much shortwave radiation to pass through the clouds and not reflecting enough shortwave radiation out to space. This corresponds to too much shortwave radiation reaching the surface.'

*Line 378. Perhaps note that for subpolar ML, StC, MS SHAP, CFL is larger than LWP, suggesting cloud amount rather than in-cloud LWP is the dominant source of bias for these cloud type?? If yes, shouldn't this be reflecting in the main conclusion. Its LWP overall, but for some cloud types CFI is the dominant bias?*

We have now included this more explicitly in the results and conclusions.

Line 437: 'For the sub-polar and mid-latitude regions, lower LWP biases correspond to lower SHAP values for these same cloud types, despite their continued dominance (indicated by size), indicating that for these regions, these optically thick, mid-low level clouds are not driving the $SWCRE_{TOA}$ bias, but rather the CFL is.'

Line 591: 'In addition, we have shown that there are cloud-type specific behaviours that can be easily captured using this type of analysis, including that for mid-latitude and sup-polar mid-level, stratocumulus and marine stratiform clouds the CFL has higher SHAP values than the LWP, indicating a greater importance.'

*Line 394. This seems like pretty weak speculation to me. Perhaps add "we speculate …". Rather to me it seems likely that CTP is to a large degree taking the role of providing measure of cloud-type. (Perhaps offer this as an alternative speculation?)*

That is a good point. We have added this as an explanation instead.

Line 405: 'We speculate that this field is providing some measure of the cloud type, which is supported by the strongly linear relationship with the cloud types derived previously (with total cloud optical depth and cloud top pressure).'

*Line 422. The noise is obvious, but it is not obvious to me that the noise is "introduced by the nudging". Do you mean the signal is small (because nudging limits the effect of the change in ice capacitance as compared to a free running model) and so the "signal to noise" is small? Or do you mean that some the "noise", that is the variation in the fields examined is actually larger than would be the case for a free running model? Please clarify.*

In a previous study, (Fiddes et al. 2021), which compared perturbed nudged simulations to similarly perturbed free-running simulations, we saw that the nudged simulations had a smaller response, with a lot more seemingly random variation, as seen in this study. So in that respect, we conclude that this 'noise' is introduced by the nudging, but the exact mechanism how, we have not identified. We have included some extra description of this in an earlier paragraph:

Line 480: 'This study compared perturbed nudged simulations to perturbed free-running simulations. The nudged simulations had a smaller overall response, with a lot of seemingly random variation, which, as in this study, we are referring to as 'noise'. '

*Line 438. You write "For CFL, higher clouds (cirrus, convective, frontal) are found to be associated with a reduction in CFL and an increase in SHAP values." Any idea why ?*

Line 505: 'In the case of mid-latitude convective clouds we suggest this is caused by the associated increase in LWP converting into an increase in rainfall, overall reducing cloud lifetime. For the remaining cloud types, the picture is less clear with very little change in rainfall.'

*Line 481. Perhaps changes "like-for-like" to "coincident in time"? (To me just using a satellite similar means compare like-to-like")*

We have made this change as suggested.

*Line 489. While I am sure it is true there isn't one "fix", that the radiation bias is asymmetric does not demonstrated this. Rather I would argue that different cloud regimes have different biases, and there are differences in the distribution of cloud regimes in different locations.*

Good point, we have revised this statement.

Line 567: 'This asymmetry has not previously been considered, and is possibly a reflection of the differences in SWCRE$_{TOA}$ biases between cloud regimes, or other unaccounted for physical processes.'

*Line 491. Perhaps change "cannot be" to "the XGBoost model suggest that biases cannot be explained by ... ".*

We have revised this sentence as suggested:

Line 569: 'Importantly, the XGBoost model suggests that the ACCESS-AM2 SO SWCRE$_{TOA}$ bias cannot be completely explained by ...'